# Vacuolar-type H⁺-ATPase-mediated extra-organellar buffering resolves mitochondrial dysfunction

Geoffray Monteuuis[1], Ryan Awadhpersad [1], Daan van der Kolk[1], Sachin K. Singh[2], Tuula A. Nyman [2], Alina Malyutina[3,4], Nicola Zamboni [5], Kari Moisio[6,7], Juhana Juutila[6,7], Ville Hietakangas [6,7], Sara Seneca[8], Christopher J. Carroll [9] & Christopher B. Jackson [1] ✉

Mitochondrial dysfunction underlies a wide range of human diseases, including primary mitochondrial disorders, neurodegeneration, cancer, and ageing. To preserve cellular homeostasis, organisms have evolved adaptive mechanisms that coordinate nuclear and mitochondrial gene expression. Here, we use genome-wide CRISPR knockout screening to identify cell fitness pathways that support survival under impaired mitochondrial protein synthesis. The strongest suppressor of aberrant mitochondrial translation defects – besides a compendium of known mitochondrial translation quality control factors – is the loss of the vacuolar-type H⁺-ATPase (v-ATPase), a key regulator of intracellular acidification, nutrient sensing, and growth signaling. We show that partial v-ATPase loss reciprocally modulates mitochondrial membrane potential ($\Delta\Psi_m$) and cristae structure in both cancer cell lines and mitochondrial disease patient-derived models. Our findings uncover an extra-organellar buffering mechanism whereby partial v-ATPase inhibition mitigates mitochondrial dysfunction by altering pH homeostasis and driving metabolic rewiring as a protective response that promotes cell fitness.

Essential energy and biosynthetic pathways converge at mitochondria. Hence, the impairment of mitochondrial function by genetic, environmental, or age-related factors has broad implications in human disease, including neurodegenerative, cardiovascular and metabolic diseases, as well as cancer and ageing[1,2]. The spectrum of diseases involving mitochondrial dysfunction is a consequence of the multiple roles of mitochondria in metabolic and oxidative phosphorylation system (OXPHOS)-related processes[3]. Mitochondria are abundant double-membrane compartmentalized organelles within the

mammalian cell harboring a streamlined multi-copy circular genome[4,5]. As a remnant of its prokaryotic ancestor, mammalian mitochondrial DNA encodes 13 highly hydrophobic subunits of the OXPHOS system alongside 22 transfer RNAs (tRNA) adapted for mitochondrial codon usage and 2 ribosomal RNAs (rRNA); structural components of the specialized mitochondrial ribosomes (mitoribosomes). An estimated ~1100 nuclear-encoded proteins are required for mitochondria-specific pathways in a tissue- and cell-type specific manner, with nuclear and mitochondrial OXPHOS subunits

[1]Department of Biochemistry and Developmental Biology, Faculty of Medicine, University of Helsinki, Helsinki, Finland. [2]Department of Immunology, Institute of Clinical Medicine, University of Oslo and Oslo University Hospital, Oslo, Norway. [3]Research Program in Systems Oncology, Faculty of Medicine, University of Helsinki, Helsinki, Finland. [4]Institute for Molecular Medicine Finland, HiLIFE, iCAN Digital Precision Cancer Medicine Flagship, University of Helsinki, Helsinki, Finland. [5]Institute of Molecular Systems Biology, Department of Biology, ETH Zurich, Zurich, Switzerland. [6]Faculty of Biological and Environmental Sciences, University of Helsinki, Helsinki, Finland. [7]Institute of Biotechnology, HiLIFE, University of Helsinki, Helsinki, Finland. [8]Center for Medical Genetics/ Research Center Reproduction and Genetics, Universitair Ziekenhuis Brussel, Brussels, Belgium. [9]Genetics Section, Cardiovascular and Genomics Research Institute, City St. George's, University of London, London, UK. ✉e-mail: christopher.jackson@helsinki.fi

requiring strict coordination and stoichiometric expression[6]. Subsequently, quality control at the level of mitochondrial translation represents an important hub at which mitochondria orchestrate proteostatic and metabolic stress responses[7]. In particular, due to their prokaryotic ancestry, mitoribosomes are unintended targets of antibiotics and, owing to their ability to promote mitochondrial function, an interesting pharmacological target for cancerous metabolism[8].

Genome-scale approaches have emerged as a powerful tool to identify essential and cellular fitness pathways[9,10], with several recent studies focusing on uncovering new aspects of mitochondrial biology[6,11–15]. These approaches encompassed OXPHOS-dependence[12], small-molecule mitochondrial inhibitors[13], DELE1-dependent cytosolic relaying upon mitochondrial dysfunction[11,14], and co-regulators of mitonuclear expression[15].

Here, we pharmacologically targeted the mitoribosome to induce aberrant mitochondrial translation (mistranslation) resulting in $\Delta\Psi_m$ ablation and inhibit mitochondrial translation (nontranslation) resulting in subsequent loss of OXPHOS complexes, in order to identify pathways increasing cell fitness. To explore mitochondrial quality control mechanisms elicited under perturbed mitochondrial proteostasis, we used genome-wide CRISPR/Cas9 knockout screening in combination with pharmacological targeting of mitoribosomes to uncover cellular adaptations and elicited rescue pathways and map a compendium of mitochondrial translation interactors.

Importantly, we identify the vacuolar (v-)ATPase as a reciprocal regulator of mitochondrial morphology and mitochondrial membrane potential ($\Delta\Psi_m$). This unique organellar adaptation alleviates mitochondrial dysfunction and recovers the $\Delta\Psi_m$ by regulating pH homeostasis and rewiring of metabolic pathways. This fundamental relationship implies that partial v-ATPase inhibition might be beneficial in specific conditions of mitochondrial dysfunction with relevance beyond primary mitochondrial disorders, including neurodegenerative diseases (e.g., lysosomal storage diseases), cancer and ageing.

## Results

### Genome-wide screening identifies suppressors of mitochondrial mis- and nontranslation

To uncover genetic pathways suppressing mitochondrial dysfunction, we applied a genome-wide CRISPR screen using growth as a fitness indicator in cells treated with small molecules targeting mitochondrial translation (Fig. 1A). We hypothesized that mitochondrial mistranslation-induced cell arrest is suppressed by loss of genes regulating mitochondrial translation. To induce mitochondrial mistranslation, we used the peptide mimetic actinonin (ACT) resulting in the accumulation of mistranslated, membrane-inserted mitochondrial proteins causing subsequent membrane potential loss[16–21]. ACT-induced mistranslation is fully suppressed by complete inhibition of mitochondrial translation using chloramphenicol (CAP)[17,18] (Figs. 1B and S1A). CAP acts on the large ribosomal subunit (LSU) by inhibition of peptide bond formation, thereby stalling mitoribosomes resulting in loss of mitochondrial translation with subsequent loss of mitochondrial-encoded respiratory chain complexes (Figs. 1B and S1A). Both compounds act on the mitochondrial ribosome, yet have a dissimilar effect on mitochondrial structure, induced stress responses, and subsequent cellular metabolic remodeling. CAP reduces OXPHOS complexes, whilst ACT leads to stress-induced mitochondrial dynamin-like GTPase OPA1 processing and subsequent growth arrest[21,22]. The resulting accumulation of shorter soluble OPA1 isoforms (S3-5) is indicative of mitochondrial fragmentation (Figs. 1B and S1A). Prolonged treatment with CAP induces a different processing of OPA1 with reduction of L1 and accumulation of S4 isoforms (Figs. 1B and S1A). To investigate how dysfunctional mitochondrial protein synthesis affects mitochondrial-nuclear expression, we determined the minimal ACT and CAP concentrations having the maximal effect on mitochondrial translation and inducing the greatest mitochondrial-nuclear imbalance (indicated by the MT-CO2/SDHA ratio, Figs. 1B and S1A, B). These concentrations permitted residual cell growth sufficient for 8-12 population doublings to complete the screen. At 15 μM ACT, there is prominent OPA1-processing whilst mitochondrially translated subunits are moderately reduced compared to untreated condition (Figs. 1B and S1A). Overall, the steady-state levels of OXPHOS proteins decreased in all conditions, with the least in ACT (Figs. 1B and S1A, C). Mitochondrial respiration was impaired in all CAP-containing conditions where MT-CO2 protein levels were decreased as a consequence of loss of mitochondrial protein synthesis (Fig. S1D). As a readout of mitochondrial dysfunction, $\Delta\Psi_m$ was decreased in all screen conditions, most significantly when treated with ACT only (Fig. S1E). Decreased mitochondrial ribosomal protein (MRP) levels (Figs. 1B and S1A) in ACT suggested mitochondrial ribosome instability, which we confirmed using label-free quantitative proteomics (Fig. S1F). The cytoplasmic ribosome was differentially affected in the screen condition with ACT only, showing a significant decrease in abundance of cytoplasmic ribosomal subunits, possibly associated with growth arrest (Fig. S1F). The loss of mitochondrial translation using 5 μg/mL CAP, however, suppressed ACT-induced mistranslation and rescued mitoribosome decay, OPA1 processing and $\Delta\Psi_m$ to CAP levels (Figs. 1B and S1A, E[17]). Mitochondrial DNA (mtDNA) copy number was not significantly altered in either of the conditions (Fig. S1G). All conditions showed oxidative phosphorylation and mitochondrial dysfunction as highly enriched in pathway analysis, with a distinct mechanism of affecting mitochondrial translation (Fig. S1H, I).

The genome-wide CRISPR screen was performed in HEK293 cells using the Brunello library[23] encompassing ~77,000 guide RNAs (sgRNAs) against ~19,000 human genes (4 sgRNAs per gene) (Fig. 1C). All drug concentrations and the ACT CAP combination were adopted to lead to a comparable growth inhibition at low dose (Fig. 1D). This was performed to avoid subsequent proliferation-induced hit biases - prominent with loss of mitochondrial genes - when comparing conditions[24,25]. The resulting data showed glycolysis-associated genes to be synthetic sick in all conditions, demonstrating mitochondrial deficiency to persist during the screen[24,25], allowing us to define a unique synthetic sick gene signature encompassing glycolysis pathway genes (e.g., *SLC2A1/GLUT1*, Figs. 1E and S2A). As expected, the combination of both drugs (ACT CAP condition) showed CAP to blunt both suppressing and synthetic sick gene signatures of ACT (Fig. 1E). Conclusively, induction of mistranslation and subsequent inner mitochondrial membrane stress through ACT identified several RNA-processing enzymes (e.g., *FASTKD* family members) and cristae morphology regulation (*YME1L1*) to be essential, and the loss of genes involved in mitochondrial translation (e.g., *C6ORF203* and *REXO2*) as suppressors. This allowed us to define a compendium of proteins mediating mitochondrial translation as a mechanism to resolve mistranslation stress and hence genes involved in mitochondrial quality control (Fig. 1F). Of the top 50 enriched genes, 13 were associated with mitochondrial translation directly or via RNA-processing steps, of which three were mitoribosomal proteins (Fig. 1F). The highest enriched genes involved in mitochondrial translation were the RNA-binding protein *C6ORF203/MTRES1* and *METTL17*, the rRNA-modifying enzymes *GTPBP3*, functionally related *GTPBP8*, the oligoribonuclease *REXO2* and the mitochondrial release factor *C12ORF65/MTRFR* (Fig. 1F). Synthetic lethal genes unique to ACT encompass *FASTKD1*, *PTCD2*, and *MDH2* (Fig. 1F). Knockout of candidates *C6ORF203/MTRES1* or *MRPL44*, whose loss blunt mitochondrial translation, showed insensitivity to ACT-induced growth arrest confirming the validity of the results of the screen and specificity of ACT (Fig. 1G). Within annotated mitochondrial proteins (MitoCarta3.0[26]), as expected, mitoribosomal subunits formed a cluster (Fig. S2B)[26]. Aberrant mitochondrial translation (mistranslation) and subsequent loss of $\Delta\Psi_m$ therefore rendered functional OXPHOS complexes essential (Fig. S2C).

Unexpectedly, suppressor genes of mitochondrial mistranslation included multiple subunits of the v-type H⁺-ATPase, which was the sole

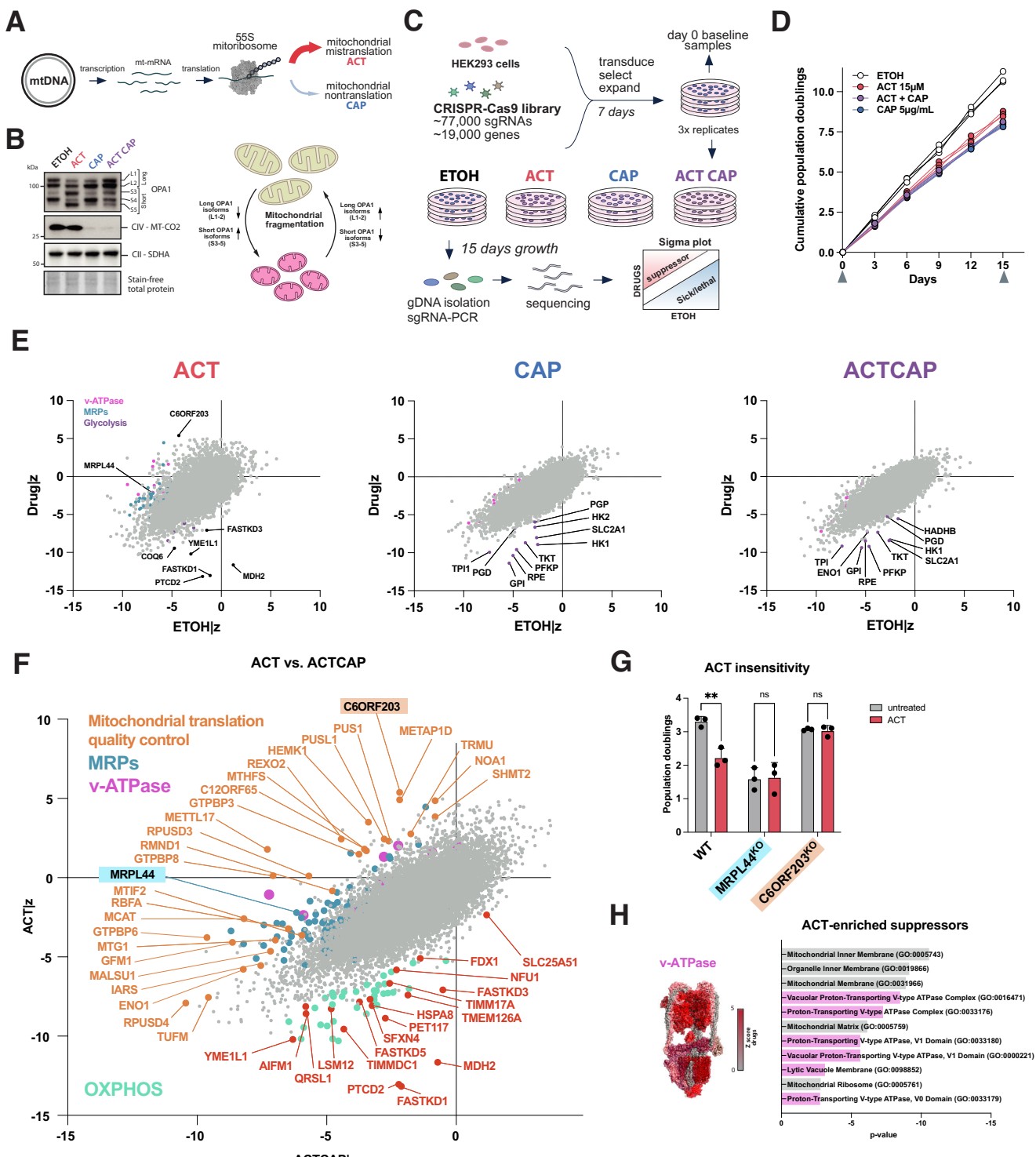

significant extra-mitochondrial suppressor (Fig. 1E, F, H). In conclusion, along a compendium of genes of mitochondrial translation-related quality control and proteostasis factors, we identified subunits of the extra-mitochondrial vacuolar (v-)ATPase as the strongest suppressors of mitochondrial mistranslation.

**Loss of the vacuolar (v-)ATPase is a potent suppressor of mitochondrial translation**

From our screen, among the strongest suppressors of inner mitochondrial membrane stress were genes encoding for subunits of the H⁺-type vacuolar (v-)ATPase (Fig. 2A). v-ATPases are multi-subunit rotary proton-pumping complexes involved in the degradation and sorting of macromolecules, nutrient storage and release, affecting intracellular pH and ion homeostasis[27,28]. The genes with the highest cell fitness increase were with loss of v-ATPase subunits, located in the ATP-hydrolyzing peripheral $V_1$ domain and integral proton-pumping $V_0$ domain (Fig. 2B). The strongest suppression was related to the ATPase H⁺-transporting accessory protein 1 (*ATP6AP1/Ac45*), for which we generated a knockout (KO) in HEK293 cells (Fig. 2C). Knockout of *ATP6AP1* did not affect the stability of the $V_1$ subunit v-ATPase A1 or v- ATPase B1/2 (Fig. 2C). ATP6AP1^KO cells were highly sensitive to the v-ATPase inhibitors bafilomycin A1 (bafA) and concanamycin A (concA) at low

**Fig. 1 | Genome-wide CRISPR screening identifies genes that increase cell fitness under mitochondrial dysfunction caused by mitochondrial mis- or non-translation. A** Mitochondrial translation inhibitors and specific mechanism of action on the mitochondrial ribosome. Actinonin (ACT) induces mitochondrial mistranslation; chloramphenicol (CAP) inhibits nascent chain synthesis leading to mitochondrial nontranslation. **B** Left: Immunoblot for mistranslation-inducing ACT, nontranslation-inducing CAP and their combination from samples collected at day 15 of the CRISPR screen. Mitochondrially-translated MT-CO1 and cytoplasmically-translated SDHA, mistranslation-induced OPA1 processing is suppressed by CAP-induced nontranslation in HEK293 cells. Right: OPA1 processing in relation to mitochondrial fragmentation. Representative image from one of three independent experiments. **C** Schematic outline of the CRISPR screen. **D** Low-dose-adapted population doublings under single or combined drug treatment over 15 days; three independent replicates per condition. Arrowheads indicate sampling points. **E** Sigma (Z) plots of mis- and non-translation. Gene-level scores were obtained by z-scoring sgRNA $\log_2$ fold-changes within each condition and averaging across 4 sgRNAs per gene; the relative difference of EtOH|z to Drug|z is plotted as a single dot per gene. **F** Sigma (Z) plot of the relative difference of ACT|z to ACTCAP|z identifies the compendium of suppressors of mitochondrial translational quality control (orange), including expected MRP subunits (blue) and specific synthetic sick genes associated mostly associated with IMM (red) and maintenance of OXPHOS complexes (turquoise). Extra-mitochondrial suppressor genes relating to v-ATPase in pink. **G** Drug insensitivity in growth phenotypes of C6ORF203[KO] and MRPL44[KO] to confirm suppressor screen candidates of mitochondrial mistranslation; 3 days treatment. Data represent the mean ± SD of three independent biological replicates (dots), analyzed using a 2-way ANOVA. **H** Pathway analysis of suppressors of mitochondrial mistranslation. GO enrichment was performed on ACT-enriched suppressors (Z-score difference ≥ 3, ACT vs ACTCAP); $p$ values were computed by right-tailed Fisher's exact test. Statistical significance: ns not significant, *$p \leq 0.05$, **$p \leq 0.01$. ***$p \leq 0.001$, ****$p \leq 0.0001$. ACT actinonin, CAP chloramphenicol, ETOH ethanol solvent control, MRP mitochondrial ribosomal protein.

concentrations not affecting cell growth in WT cells (Fig. 2D), suggesting impaired but residual v-ATPase function, essential for cellular viability[29]. To assert ATP6AP1 as a suppressor of mistranslation stress, we exposed both WT and knockout to increasing concentrations of ACT. Untreated ATP6AP1[KO] cells showed minimally impaired growth compared to WT cells but were completely insensitive to ACT-induced growth arrest at 15 μM (Fig. 2E). We probed for OPA1 isoforms to investigate whether the ACT-insensitivity was related to OPA1-processing, which correlates with mitochondrial fragmentation. Whilst ACT did induce OPA1-processing in WT cells, we detected an increased abundance of the long OPA1 isoforms L1/L2 upon loss of ATP6AP1 (Figs. 2E and S3A), with the YME1L1-dependent S3 short OPA1 isoform[30–32] being reduced. In line with remodeling of the cristae, the MICOS complex cardiolipin-interacting protein APOOL/MIC27 was increased while the major subunit MIC60/Mitofilin remained unaffected (Fig. S3A). The OPA1-processing inner membrane protease YME1L1, whose loss in the screen condition of mistranslation was identified as synthetic sick, remained unaltered in ATP6AP1[KO] cells treated with ACT[30–32] (Fig. S3A). Suppression of ACT-induced mistranslation by CAP did not alter YME1L1 or MIC27/APOOL levels in 15-days of treatment (Fig. S3B). Furthermore, we observed no significant effect on mtDNA copy number in the knockout or treatment, consistent with unchanged MIC60 expression levels, which is known to play a role in nucleoid organization (Fig. S1C[33]). The ratio of OPA1 isoforms correlates with mitochondrial structure, where an increase of short isoforms is associated with mitochondrial fragmentation (Fig. S3D). Transient overexpression of FLAG-tagged ATP6AP1 reduced the abundance of long OPA1 isoforms in both WT and ATP6AP1[KO] cells treated with ACT, consistent with the opposing trend observed upon ATP6AP1 loss[33]. Staining for the outer mitochondrial membrane protein TOM20 to assess mitochondrial structure revealed increased fragmentation in ACT-treated WT cells, whilst ATP6AP1[KO] were unaltered (Figs. 2F and S3F). Consistent with this, the loss of ATP6AP1 did reverse the ACT-induced deficiency of native OXPHOS complexes (Fig. S3G) and restored complex IV-impaired respiration (Fig. S3H).

Mistranslation stress leading to the most significant decrease in $\Delta\Psi_m$ amongst our screen conditions (Fig. S1E) prompted us to assess changes in $\Delta\Psi_m$ in our partial v-ATPase-deficient model. ATP6AP1[KO] cells exhibited an increased $\Delta\Psi_m$ compared to WT cells, which was further elevated by ACT treatment (Figs. 2G and S3I). The increase in $\Delta\Psi_m$ in ATP6AP1[KO] cells may explain their improved capacity to handle mistranslated proteins and mitigate inner mitochondrial membrane stress. Consistent with this, ATP6AP1[KO] cells are more resistant to the ionophore FCCP, which dissipates $\Delta\Psi_m$ (Fig. S4A). The effects of potassium ionophore valinomycin[34], the non-selective kinase inhibitor staurosporine[35] and $F_0F_1$-ATPase inhibitor oligomycin, all directly or indirectly associated with $\Delta\Psi_m$ hyper- or depolarization, were suppressed in the ATP6AP1[KO] cells (Fig. S4B–D).

To understand the effect of inner mitochondrial membrane perturbation in our conditions, we assessed the cristae organization by electron microscopy. ACT-treated WT cells showed extensive fragmentation and loss of inner membrane organization. The partial loss of v-ATPase in combination with ACT restored inner mitochondrial membrane sheets in the ATP6AP1[KO] cells (Fig. 2H), suggesting this WT-like appearance was due to the normalized long to short OPA1 isoform ratio (Fig. 2I). To test whether the increase in cell fitness with partial loss of v-ATPase was only related to the ACT-induced growth arrest, we treated ATP6AP1[KO] cells with increasing levels of CAP and ethidium bromide (EtBr) to mimic loss of mitochondrial translation. This revealed an increased resistance to CAP and EtBr in the ATP6AP1[KO] cells, suggesting a common mechanism enhancing cell fitness in response to mitochondrial dysfunction (Fig. S4E, F).

In conclusion, we determine the function of the extra-mitochondrial v-ATPase complex as a negative regulator of OPA1-processing and $\Delta\Psi_m$, influencing cellular resistance to mitochondrial dysfunction.

## ATP6AP1 associates with the $V_0$ sector of lysosomal v-ATPase, and its loss impairs lysosomal acidification

v-ATPases are present in multiple cellular locations as proton pumps, such as the plasma membrane, Golgi-derived and secretory vesicles, endosomes and lysosomes[27]. To determine the specific location, we used transiently expressed FLAG-tagged ATP6AP1 as bait, which localized almost exclusively into discrete vesicles throughout the cytosol (Fig. 3A). Subcellular fractionation experiments showed that ATP6AP1 co-migrates and is enriched in the same fraction as the lysosomal protein LAMP1 rather than the plasma membrane glucose transporter GLUT1 (Fig. S5A).

The ms/ms analysis using ATP6AP1-FLAG as bait detected $V_0$ domain subunits at similar levels in immunoprecipitates from cells treated with or without ACT (Fig. 3B). The analysis of other proteins captured with ATP6AP1-FLAG as bait were associated predominantly with the lysosome (Fig. 3C, D). These findings prompted us to investigate the pH of intracellular compartments. To assess how the loss of ATP6AP1[KO] affected acidic compartments, we used the hydrophobic dye LysoTracker Red DND-99, which in acidic environments becomes rapidly protonated, predominantly labeling late endosomes and lysosomes. Both ACT-treatment and ATP6AP1[KO] cells displayed an increase in size of acidic compartments, also reflected in acridine orange staining (Figs. 3E and S5B, C). Due to the increased size of acidic vesicles, we stably expressed the ratiometric biosensor mTFP1-LAMP1-mCherry (FIRE-pHLy[36]) based on luminal proton-quenchable TFP1 and cytosolic-facing mCherry to assess lysosomal pH (Figs. 3F and S5D). The WT cells had a lysosomal pH of ≈4.4, which increased to ≈4.7 upon ACT treatment. The ATP6AP1[KO] showed an increased lysosomal pH of ≈4.8, which was unaltered upon ACT treatment. Lysosome-associated membrane glycoprotein 1 and 2 (LAMP1/2) showed comparable levels

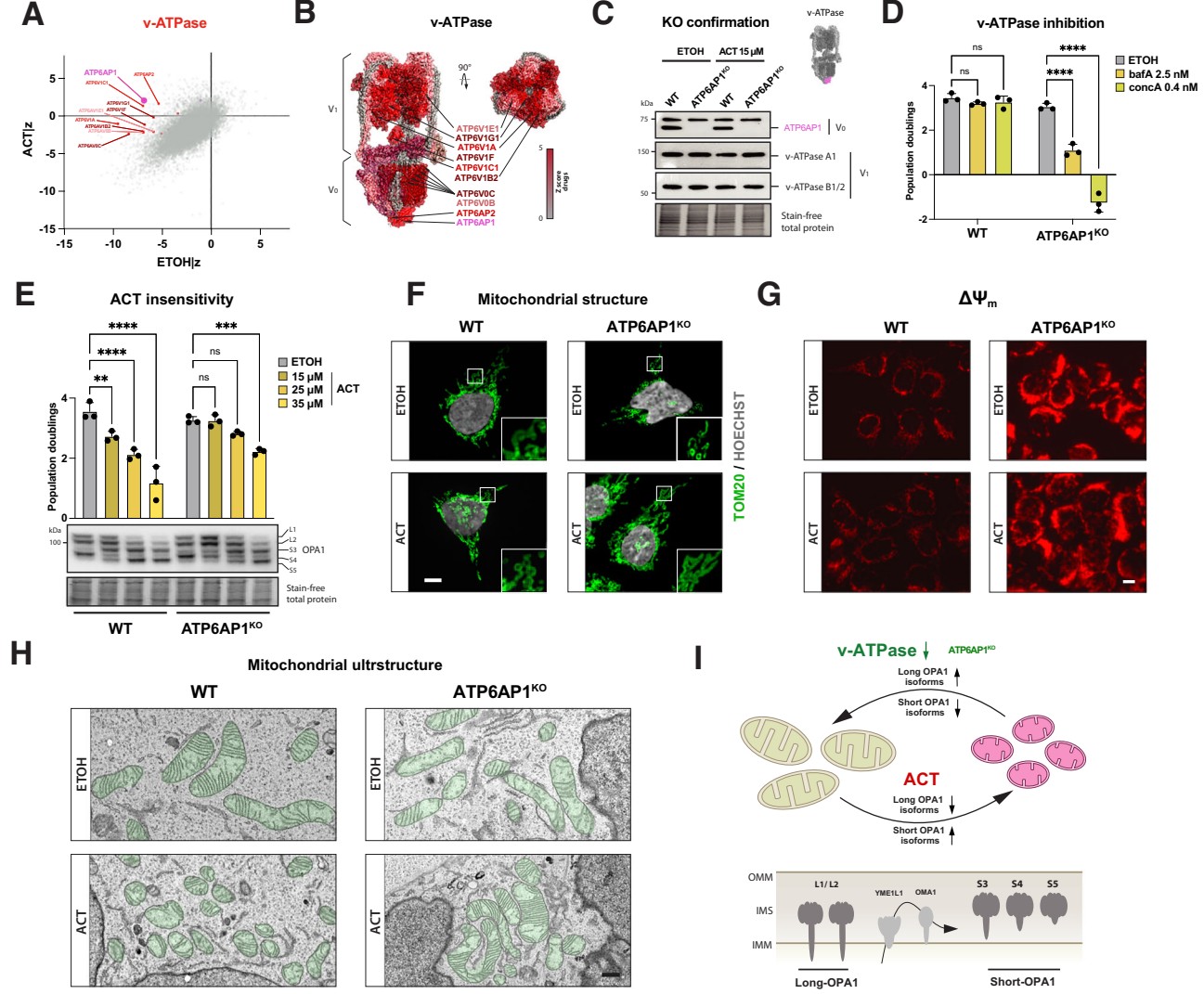

**Fig. 2 | Partial loss of vacuolar (v-)ATPase acts as a suppressor of mitochondrial dysfunction and increases cell fitness. A** Sigma plot of screen condition for mistranslation, relative difference of EtOH|z to ACT|z, v-ATPase subunits highlighted. **B** v-ATPase structure with color-coded drug suppressor $Z$-scores. PDB structure: 6WM2, v-ATPase state 1 with ADP, SidK hidden. **C** Immunoblotting for v-ATPase subunit knockout (ATP6AP1$^{KO}$) confirmation and probing for steady-state level of v-ATPase V1, v-ATPase B1/B2. **D** v-ATPase subunit knockout (ATP6AP1$^{KO}$); drug-sensitivity of v-ATPase inhibitors bafA and concA. **E** Growth curves and representative immunoblot of OPA1 isoforms to assess drug insensitivity and mitochondrial fragmentation status. Data represent the mean ± SD of three independent experiments (dots), 2-way ANOVA. **E** Growth curves to assess drug sensitivity and representative immunoblot of OPA1 isoforms as proxy of mitochondrial fragmentation. Data represent the mean ± SD of three independent experiments

(dots), 2-way ANOVA. **F** Immunofluorescence microscopy assessing mitochondrial structure via outer mitochondrial membrane (OMM) using TOM20. DNA stained with Hoechst. Scale bar: 2 µm. **G** ΔΨm assessed via TMRM fluorescence intensity in WT and ATP6AP1$^{KO}$ cells. Scale bar: 2 µm. **H** Electron microscopic analysis of WT and ATP6AP1$^{KO}$ cells. Mitochondria are shaded in green. Scale bar: 500 nm. **I** Schematic of OPA1-processing and extra-mitochondrial reversal of mitochondrial fragmentation. All treatments performed in HEK293 cells for 3 days if not indicated otherwise. Bars plots represent the mean ± SD of three independent experiments (dots), analyzed using a 2-way ANOVA with ns not significant, $*p \leq 0.05$, $**p \leq 0.01$, $***p \leq 0.001$, $****p \leq 0.0001$. ACT actinonin, Bafilomycin A1 bafA, concanamycin A concA, ETOH ethanol solvent control, IMM inner mitochondrial membrane, IMS intermembrane space, OMM outer mitochondrial membrane, TOM20 Translocase of outer mitochondrial membrane 20.

but were retained higher in the gel in ATP6AP1$^{KO}$ cells, potentially caused by altered glycosylation (Fig. S5E). Light chain 3B (LC3B) lipidation ratio indicating autophagic turnover was unaltered across all conditions (Fig. S5F). Complete inhibition of v-ATPase-dependent lysosomal acidification is associated with cellular iron deficiency and subsequent mitochondrial dysfunction affecting mitochondrial-iron-sulfur cluster integration into OXPHOS complexes[37]. However, the ATP6AP1$^{KO}$ cells showed no iron-deficiency-induced Hif1α activation, nor were they more susceptible to deferoxamine-induced iron depletion (Fig. S5G).

Next, we investigated the effect of long-term low v-ATPase deficiency on cytosolic pH, given that alterations in cytosolic pH

homeostasis can have significant consequences for mitochondrial function—and vice versa. Notably, cytosolic acidification has been reported in models of mitochondrial dysfunction, particularly those associated with loss of $\Delta\Psi_m$[34,38–40]. Using the pH-calibrated ratiometric probe BCECF AM, we observe a non-significant decrease under ACT-treatment, whereas the knockout sustains an alkaline cytosolic pH (≈0.3 units) compared to the WT, treated or untreated (Fig. 3G). Effectively, a more alkaline cytoplasmic pH reduces the ionic strength of the chemical proton gradient ($\Delta pH_m$) component of $\Delta\Psi_m$, which accounts for up to 30% of the proton gradient ($\Delta p$), driving metabolic flux for OXPHOS and activity of electroneutral ion exchangers to maintain osmolarity and volume of mitochondria[41]. To examine the

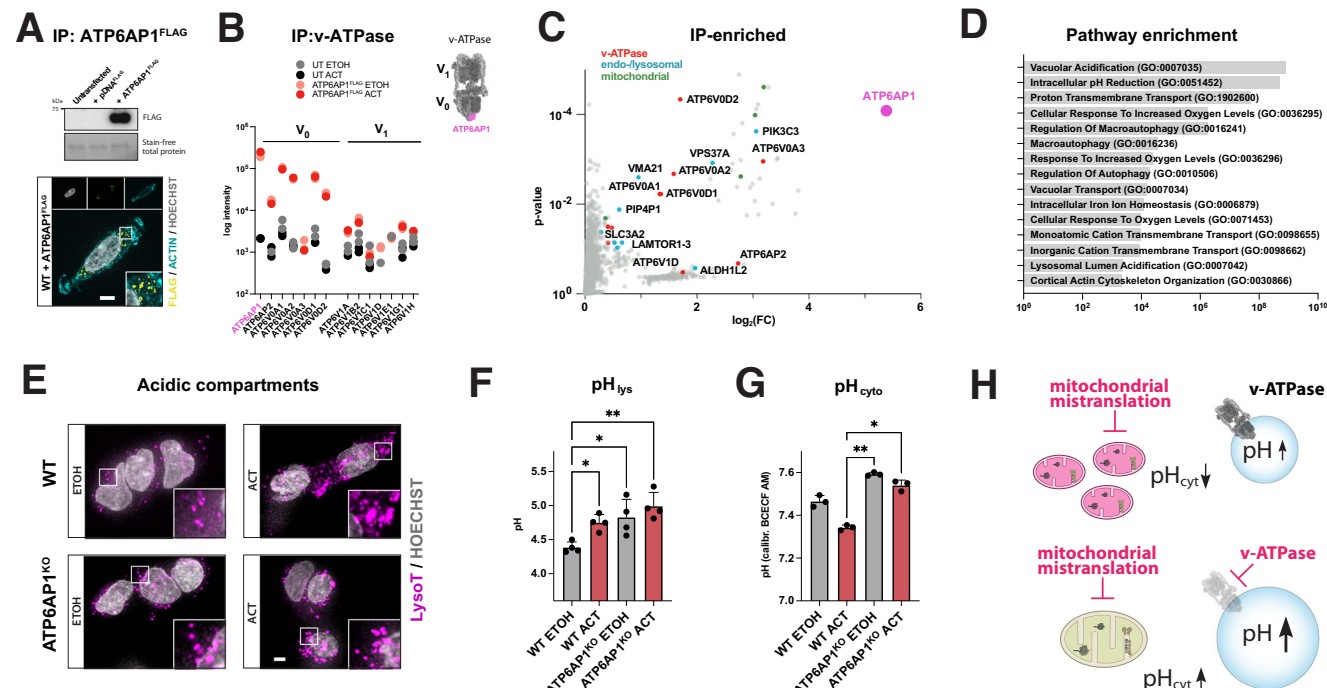

**Fig. 3 | ATP6AP1 is associated with the lysosomal v-ATPase, and its loss causes organellar acidification defects. A** IP of FLAG-tagged ATP6AP1 expression and cellular localization. DNA stained with Hoechst and the cytoskeleton with actin. Scale bar: 3 μm. **B** IP–MS-derived abundance of detected $V_0$ and $V_1$ v-ATPase subunits. Quantitative proteomics, each dot represents the mean of duplicate measurements. **C** Volcano plot of positively enriched proteins. Quantitative proteomics, dots represent the mean of two biological replicates, Student's t-test. v-ATPase (red), endo/lysosomal (cyan), and mitochondrial (green) proteins are highlighted, with selected hits labeled. **D** Positively enriched pathways from (C). *P* value cut-off >0.05; right-tailed Fisher's exact test. **E** Fluorescence microscopy

assessing acidic compartments; Lysotracker DND-99 fixed. DNA stained with Hoechst. Scale bar: 2 μm. **F** pH-calibrated determination of lysosomal acidity ($pH_{lys}$) using ratiometric FIRE-pHLy. Data represent the mean ± SD of three independent experiments (dots), analyzed using a 2-way ANOVA. **G** pH-calibrated determination of cytosolic pH ($pH_{cyt}$) using ratiometric BCECF AM. Data represent the mean ± SD of three independent experiments (dots), analyzed using a 2-way ANOVA. **H** Schematic of effects of lysosomal acidification and mitochondrial mistranslation. All treatments were performed in HEK293 cells for 3 days. Statistical significance *$p ≤ 0.05$, **$p ≤ 0.01$, ***$p ≤ 0.001$, ****$p ≤ 0.0001$. ACT actinonin, ETOH ethanol solvent control, UT untransfected.

effect of pH alterations on $\Delta\Psi_m$, we exposed WT and knockout cells to increasing acidic and alkaline media. Increased extracellular acidification or alkalinization resulted in decreased growth in both WT and ATP6AP1$^{KO}$ cells, with the latter being more sensitive at the highest doses (Fig. S5H, I). Increasing extracellular acidification led to a decrease in the $\Delta\Psi_m$ in the knockout cells only, while remaining constant in the WT cells. Conversely, extracellular alkalinization increased $\Delta\Psi_m$ in both WT and knockout cells, suggesting that mechanisms such as increased proton pumping, reduced passive proton conductance, or mitochondrial uncoupling may be compensated by a reduction in $\Delta pH_m$ (Fig. S5J, K).

In conclusion, these data show that loss of ATP6AP1 impairs lysosomal function and that changes in lysosomal pH and size of acidic compartments directly affect cytosolic pH, contributing to the resistance to mitochondrial dysfunction associated with $\Delta\Psi_m$ alterations.

### Vacuolar (v-)ATPase deficiency compromises activation of mitochondrial dysfunction-elicited biosynthetic processes and reduces mitochondrial metabolic dependence

Having established that lysosomal v-ATPase function is impaired, we next focused on the resulting metabolic consequences. v-ATPase activity is central to intracellular proton homeostasis, and its sustained deficiency can shift the proton distribution across compartments. v-ATPase-induced shifts in pH may function as a signaling cue triggering broader metabolic reprogramming, such as nutrient sensing at the lysosome, and may phenocopy a nutrient-deprived or energy-stressed state, even in the presence of external nutrients[42]. We therefore asked how lysosomal deficiency shapes central metabolic pathways. Given the impaired lysosomal acidification observed in our partial v-ATPase-

deficient model, we investigated the assembly status of the v-ATPase. Whilst the stability of $V_1$ seemed unaffected, label-free proteomics of ACT-treated and untreated ATP6AP1$^{KO}$ cells displayed decreased steady-state levels of all detected $V_0$ subunits, most prominently of the ATP6AP1-associated heterodimer-forming subunit ATP6AP2 (Fig. 4A and S6A). With the assembly and activity of the v-ATPase being responsive to glucose[28,43,44] and dysfunctional mitochondrial metabolism relying on glycolysis[45], we exposed our models to altered glucose concentrations. ATP6AP1$^{KO}$ cells remain insensitive to lower glucose availability compared to WT cells (Fig. 4B), confirming aberrant glucose-sensing potentially as a consequence of a partial loss of $V_0$. Galactose, known to force oxidative metabolism, was less well tolerated in the knockout cells but showed almost no additional growth impairment under ACT treatment as opposed to the WT. This established further that glucose becomes essential under mitochondrial dysfunction, whilst suggesting that the knockout cells are metabolically inflexible.

Next, we applied untargeted metabolic analysis to understand how v-ATPase-dependent metabolic rewiring was contributing to our observations. The most abundant metabolites in the ATP6AP1$^{KO}$ cells related to NADH, PPP-associated metabolites such as 6-phosphogluconate and ribose-5-phosphate, whilst glutamine, aspartate and L-cystathionine were decreased, also when treated (Figs. 4C and S6B). Overall, the most significantly altered pathways in ATP6AP1$^{KO}$ related to pathways elicited in the integrated stress response encompassing amino-acyl tRNA biosynthesis and amino acid and one-carbon metabolism, all of which were significantly decreased (Fig. S6C). Relative metabolic ratios determined from untargeted metabolomics show that the total ATP/AMP ratio was only altered in

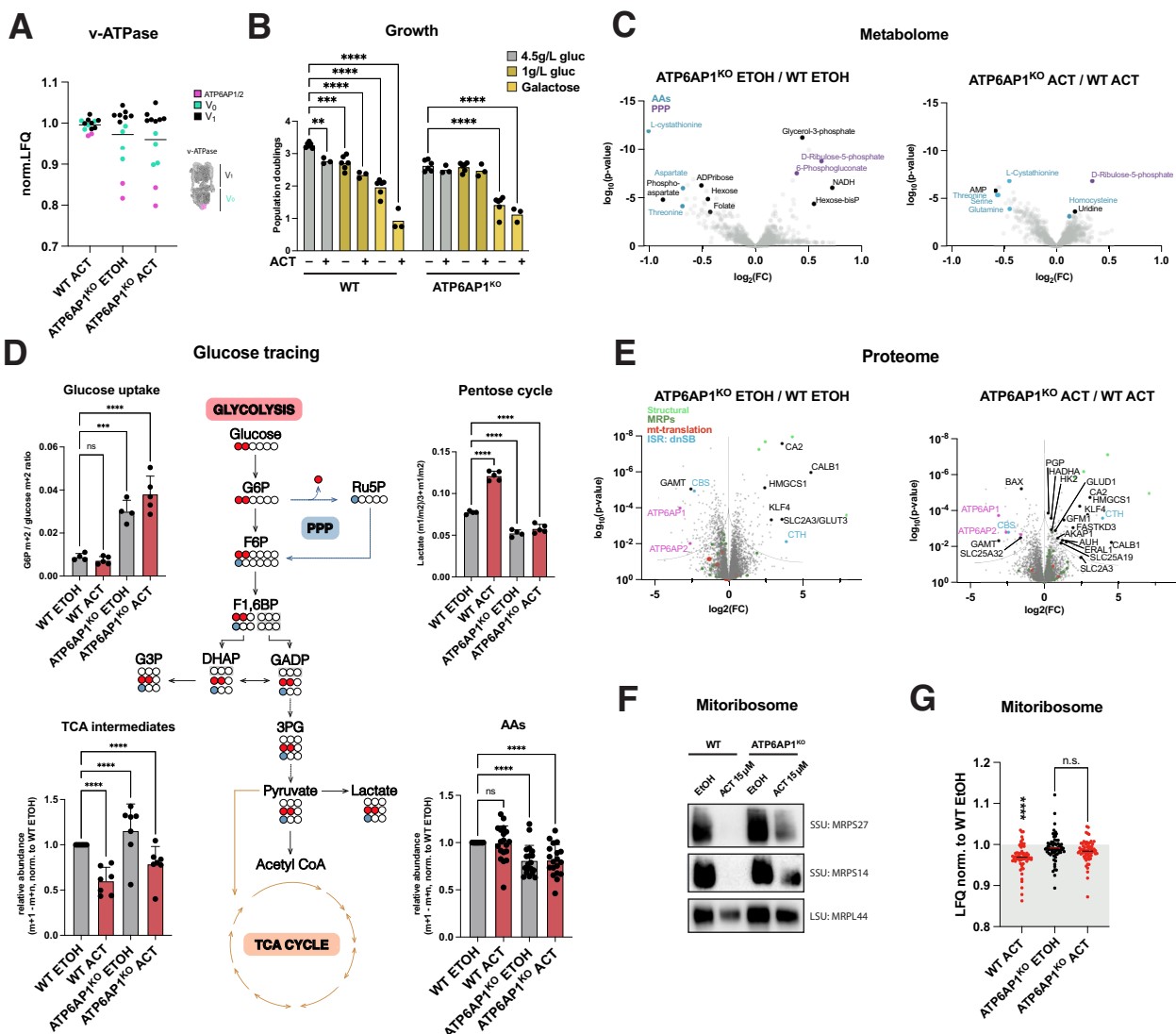

**Fig. 4 | Loss of ATP6AP1 reroutes glycolytic pathway decreasing susceptibility to mitochondrial dysfunction and attenuates integrated stress-induced pathways. A** v-ATPase subunit abundance; label-free quantitative proteomics. 15 days treatment. Per-v-ATPase-protein normalized LFQ values relative to WT ETOH.
**B** Growth phenotypes of WT and ATP6AP1[KO] in high (4.5 g/L) and low (1 g/L) glucose and galactose. Data represent the mean ± SD of independent experiments (dots), analyzed using a 2-way ANOVA. **C** Volcano plot of WT and ATP6AP1[KO]; untargeted metabolomics; each dot represents the mean of biological sextuplicates, Student's t-test. **D** Glucose tracing using [1,2-$^{13}C_2$]-labeled glucose; in upper panels each dot represents the mean of metabolite ratios derived from biological sextuplicates, in lower panels mean of biological sextuplicates were normalized to control. All bar plots show the mean with ±SD; 2-way Anova. Filled circles represent $^{13}$C-label in glucose molecules derived from glycolysis (red) or PPP combustion (blue).
**E** Volcano plot of WT and ATP6AP1[KO]; label-free quantitative proteomics, each dot

represents mean of biological quadruplicates, Student's t-test. **F** Native PAGE and immunoblot analysis of mitoribosomal assembly state WT and ATP6AP1[KO]. 25 μg total mitochondrial protein per lane. Correct detection inferred from the expected sizes of endogenous OXPHOS complexes. **G** Steady-state MRP expression levels relative to EtOH; label-free quantitative proteomics; dots represent the mean of biological quadruplicates for all detected ribosomal subunits normalized to WT ETOH. All treatments were performed in HEK293 cells for 3 days. Bars plots represent the mean ± SD of three independent experiments (dots), analyzed using a two-way ANOVA, if not indicated otherwise, with ns not significant, $*p \leq 0.05$, $**p \leq 0.01$, $***p \leq 0.001$, $****p \leq 0.0001$. 3PG 3-phopshoglycerate, AAs amino acids, DHAP dihydroxyacetone phosphate, F6P fructose-6-phosphate, F1,6BP fructose 1,6-biphosphate, G3P glycerol-3-phosphate, G6P glucose-6-phosphate; GADP gly-ceraldehyde-3-phosphate, Ru5P ribulose 5-phosphate, TCA tricarboxylic acid cycle.

ACT-treated WT, suggesting catabolic energy status alteration, whilst in response to ACT, the NAD/NADH ratio was increased in the treated ATP6AP1[KO] and GSH/GSSG or ATP/ADP remained comparable in untreated versus treated cells (Fig. S6D). The overall total cellular amino acid concentrations were altered in the ATP6AP1[KO], with serine, a key amino acid induced by the mitochondrial stress response, remaining unchanged upon ACT treatment, and cysteine, important for glutathione synthesis, depleted from the media when compared to WT cells treated with ACT (Fig. S6E). To investigate the observed glucose insensitivity and pathway rewiring in our model, we performed [1,2-$^{13}C_2$]-glucose tracing. Isotope tracing revealed increased glucose

uptake in ATP6AP1[KO] cells, as indicated by an increased glucose 6-phosphate to glucose ratio and higher incorporation of m+1 and m+2 isotopologues into glycolytic intermediates. In contrast, WT cells treated with ACT showed increased PPP cycle activity, inferred from elevated lactate m+1 to m+2 ratio labeling[46], likely due to impaired glycolytic flow and mitochondrial dysfunction (Fig. 4D). This PPP activation is suppressed in ATP6AP1[KO] cells, suggesting a metabolic rerouting of glucose away from PPP under stress conditions when v-ATPase activity is partially suppressed. Consistent with this, TCA cycle intermediate labeling patterns revealed that ACT impairs mito-chondrial glucose oxidation in WT cells, while loss of ATP6AP1 partially

restores this capacity. This metabolic shift in KO cells coincides with reduced PPP reliance and elevated glucose uptake, potentially facilitated by a more alkaline cytosolic pH. Overall, ATP6AP1[KO] cells exhibited a significant overall reduction in amino acid pools, which may result from anaplerotic flux into the TCA cycle.

Analysis of the global proteome confirmed these findings, with the mitochondrial stress-associated pathways, including de novo serine synthesis (PGHDH, PSAT1, and PHSPH) remained upregulated in the ATP6AP1[KO] cells (Figs. 4E and S6F, G)—a major pathway upregulated in all conditions of the screen (Fig. S6H), while cystathionine gamma-lyase (CTH/CSE) was substantially increased at basal level in the ATP6AP1[KO] cells, whereas it remained low in WT conditions (Fig. S6F). The prominent ribosomal decay occurring in ACT-treated WT cells was partially rescued in the ATP6AP1[KO] cells at both steady-state and native assembly (Fig. 4F, G), potentially enhancing protection from the proteotoxic stress triggered by mitoribosomal decay and the ensuing cell cycle arrest[17].

These metabolic and proteomic changes collectively indicate that in ATP6AP1[KO] cells show a metabolic phenotype protected from ACT-induced rerouting of carbons, suggestive that partial v-ATPase disruption appears to reduce cellular reliance on mitochondrial oxidation, enabling resistance to mitochondrial insults while limiting biosynthetic capacity.

### Low v-ATPase-inhibition increases $\Delta\Psi_m$ and enhances cell fitness under mitochondrial mistranslation in multiple cell-types and mitochondrial disease patient-derived fibroblasts

To confirm that the suppressed cell growth defect under ACT-induced mistranslation was due to v-ATPase deficiency, we used the highly selective v-ATPase inhibitors concA and bafA on WT cells. Titration to concentrations that did not affect cell growth resulted in multiple-fold lower concentrations than those commonly applied[47], and which would otherwise be detrimental to cell growth over a three-day treatment period (Fig. S7A). Concomitant inhibition of the v-ATPase with concA or bafA under ACT treatment improved cell fitness by restoring growth (Figs. 5A and S7B). ConcA alone did not affect OPA1 processing but showed higher L-OPA1 abundance under ACT (Fig. S7C). Acidic compartments, however, were increased with 400 pM of concA in HEK293 cells, whilst 10 nM for 1 h resulted in the expected complete loss of signal (Fig. S7D). Similarly, treatment with concA showed comparably altered metabolic pathways as in the ATP6AP1[KO] cells, including suppression of amino acid biosynthesis (Fig. S7E, F).

To establish whether the v-ATPase-dependent modulation has broader relevance, we tested the respective rescue in U2OS and primary fibroblasts. Both pharmacological inhibition of the v-ATPase or ATP6AP1 knockout in U2OS rescued growth under ACT exposure (Figs. 5B and S8A, B). Analysis of mitochondrial functional parameters in the U2OS ATP6AP1[KO] also showed higher respiration and structural preservation upon ACT treatment (Fig. S8C, D). We also observed an increase in cell fitness when primary fibroblasts were treated with ACT and concA concomitantly (Fig. 5C).

We next sought to determine whether the inhibition of v-ATPase directly effects on the $\Delta\Psi_m$ by increasing concentrations of concA over three days, which revealed a stepwise increase of $\Delta\Psi_m$ in HEK293, U2OS and fibroblasts (Fig. 5D–F). Similar to the HEK293 ATP6AP1[KO] cells, U2OS ATP6AP1[KO] displayed increased $\Delta\Psi_m$ compared to control cells, which remained unaffected upon ACT treatment (Fig. S8E). An increase in $\Delta\Psi_m$ was also observed after concA alone in fibroblasts (Fig. S7G), as well as an increase in the size of acidic compartments (Fig. S7H). Finally, combined analysis of concA-treated HEK293 and ATP6AP1[KO] in HEK293 or U2OS showed a similar metabolic rewiring (Figs. S7E and S8F).

To explore the therapeutic effect on cell fitness, we selected primary patient-derived fibroblast cell lines with confirmed mitochondrial translation defects[48–50]. All concanamycin-treated fibroblasts lines showed an increase in $\Delta\Psi_m$ and a minor increase in the size of acidic compartments (Figs. 5G and S7I). Partial v-ATPase inhibition restored their respective growth rates comparable to fibroblasts derived from healthy controls (Fig. 5H). In conclusion, partial v-ATPase inhibition increases the $\Delta\Psi_m$ and promotes cell fitness in diverse models of mitochondrial dysfunction (Fig. 5I).

## Discussion

Leveraging a genome-wide CRISPR screening strategy, we identified pathways that enhance cell fitness in response to dysfunctional mitochondrial protein synthesis. This was achieved using compounds targeting the mitochondrial ribosome, each inducing distinct stress response signatures. Under conditions of mitochondrial translation loss and OXPHOS inhibition (CAP conditions), we observed a focused essential gene set, largely restricted to glycolysis genes, which are indispensable under OXPHOS impairment. In contrast, under mitochondrial mistranslation stress, our screen identified the v-ATPase as the sole extra-mitochondrial effector mitigating mitochondrial dysfunction.

V-ATPases are critical for pH homeostasis by acidifying lysosomes and are involved in processes like macropinocytosis, autophagy, cell invasion, and cell death[27,51]. Collectively, v-ATPase activity mediates exocytosis, glucose sensing, and modulates autophagy and immune response by pH regulation[52,53]. The other non-canonical effects of the v-ATPase that are not readily attributable to its proton-pumping activity include membrane fusion, pH sensing, amino-acid-induced activation of mTORC1, and scaffolding for protein-protein interaction[42,54]. In humans, alterations in v-ATPase activity are implicated in multiple diseases, with complete loss of activity leading to embryonic lethality in higher animals[55]. In humans, ATP6AP1 deficiency is associated to N- and O-glycosylation defects resulting in immuno-deficiency hepatopathy and cognitive impairment in humans[56], whilst clonal inactivating somatic mutations in ATP6AP1/A2 are found in the majority (>70%) of all granular cell tumors [57].

While previous studies have examined the effects of strong v-ATPase inhibition, ultimately compromising cell fitness[37,58–60], our study explores the impact of partial v-ATPase inhibition in the context of mitochondrial dysfunction. In models of mitochondrial mistranslation, partial v-ATPase inhibition—achieved either through loss of the v-ATPase subunit ATP6AP1 or pharmacological inhibitors—increased cell fitness and enhanced $\Delta\Psi_m$. This inhibition benefits mitochondrial respiration and OPA1-mediated inner mitochondrial membrane architecture under mitochondrial dysfunction. To elucidate the mechanisms behind these benefits, we extensively characterized our partial v-ATPase-deficient models. ATP6AP1[KO] cells exhibited a lysosomal acidification defect, consistent with disrupted proton pumping. Interestingly, while lysosomes in ACT-treated WT cells also display lysosomal acidification impairment, the lysosomal pH in ATP6AP1[KO] cells remained unaffected by ACT treatment. Alkalinization of lysosomes—as observed in v-ATPase-deficient model—has been associated with swelling under certain physiological or cellular conditions to enhance their ability to perform critical functions. Mild lysosomal swelling, as a stress response mechanism, increases lysosomal capacity and volume, enhances tolerance to elevated waste, and functions as a proton sink by reducing proton leakage[61]. Accordingly, an increase in lysosomal volume relative to surface area, together with a relatively modest rise in pH, facilitates proton scavenging from an acidified cytoplasm. Because the v-ATPase is electrogenic, counterions must move to dissipate voltage and to facilitate bulk proton transport, further buffering pH[62]. Enlarged, less acidic lysosomes promote a shift to catabolism, altering nutrient sensing and amino acid storage to adapt metabolic flux during energy stress[63,64]. With lysosomes regulating cellular signaling pathways, swelling might help modulate these by promoting cell survival or metabolic adaptations[61,65].

Both of our ATP6AP1[KO] and v-ATPase inhibitor models show increased $\Delta\Psi_m$ attributable to multiple interdependent factors.

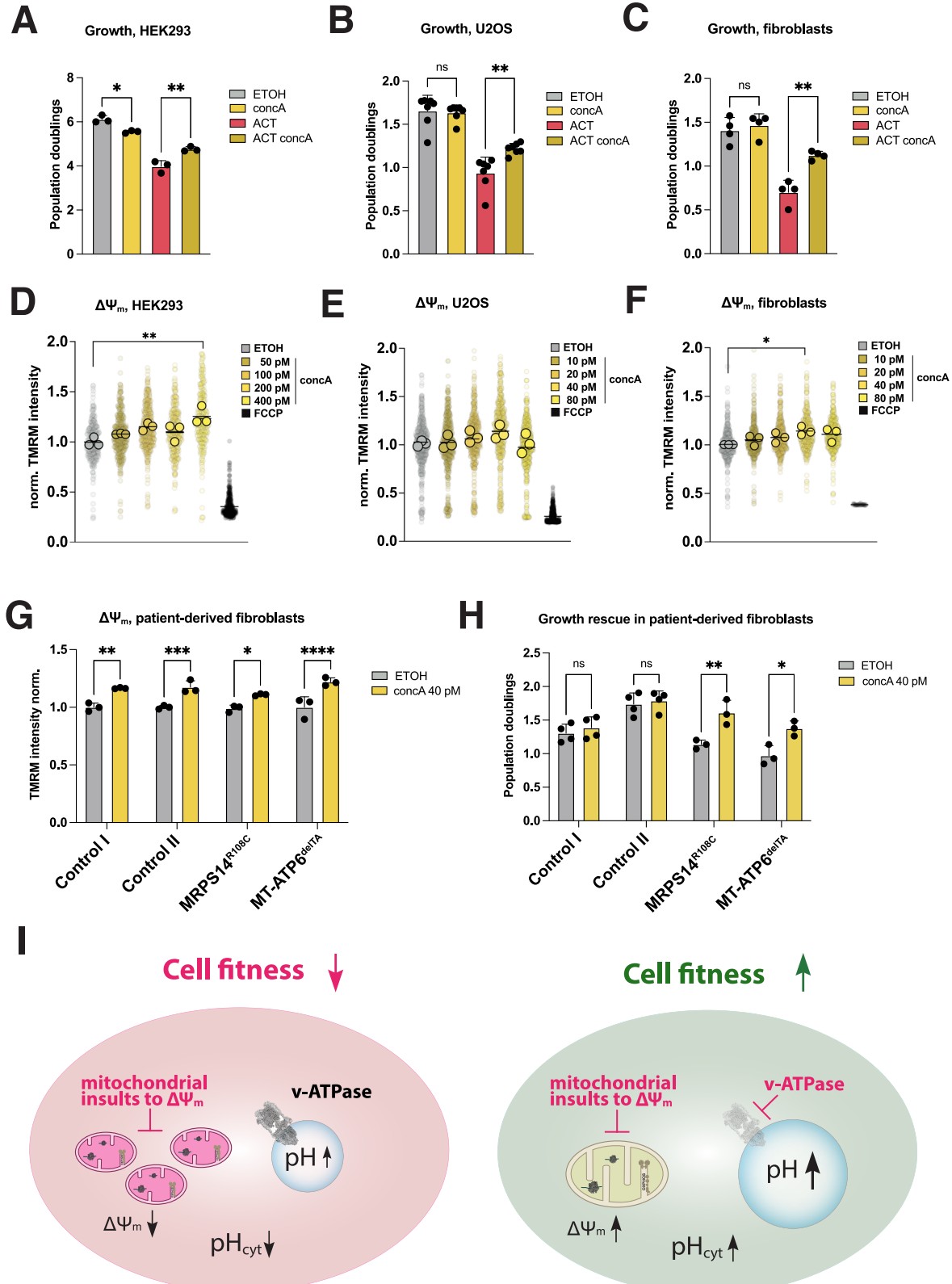

Mechanistically, an increase in $\Delta\Psi_m$ represents a compensatory mechanism to the decreased $\Delta$pH, which has been reported to reestablish the $\Delta p$ of the $\Delta\Psi_m$ through proton leakage decrease and increase in proton pumping[41,66]. Inversely, mitochondrial dysfunction has been reported to afflict lysosomes[67].

We observe that sustained low systemic v-ATPase inhibition reroutes metabolism, potentially to decrease mitochondrial dependence and promotes core essential anaplerotic pathways leading to a quiescent cell state. We also observed a slight decrease in the cytosolic pH in WT cells treated with ACT. Although not significant, this observation aligns with previous reports of decreased cytosolic pH in mitochondrial dysfunction models[34,38–40]. Interestingly, ATP6AP1[KO] cells display a more alkaline cytosolic pH at the basal level and remain unaffected by ACT treatment. As ATP6AP1[KO] cells display a higher

**Fig. 5 | Vacuolar (v-)ATPase-modulation increases $\Delta\Psi_m$ and restores cell growth in patient-derived fibroblast cell lines with mitochondrial translation defects. A–C** Growth phenotypes with concomitant pharmacological v-ATPase inhibition using concA under ACT treatment. Conditions normalized to ACT-treated WT; 6-day (HEK293) and 3-day (U2OS, fibroblasts) treatments. Dots represent independent experiments. Mean with ±SD. 2-way Anova. **D–F** $\Delta\Psi_m$ assessed via TMRM fluorescence intensity with increasing concA concentrations in WT cells. Positive control: FCCP. Small dots represent all cells measured. Large dots represent the mean of independent experiments. Mean with ±SD. 1-way Anova. **G** $\Delta\Psi_m$ assessed via TMRM fluorescence intensity in primary control and patient fibroblasts. Dots represent independent experiments. Mean with ±SD. 1-way Anova. **H** Growth rates of primary healthy control and patient-derived fibroblasts with concomitant pharmacological v-ATPase inhibition; 3 days treatment. Dots represent independent experiments. Mean with ±SD. 1-way Anova. **I** Schematic model of partial v-ATPase-deficiency mitigating mitochondrial insults reducing $\Delta\Psi_m$ and impair cell growth. All treatments performed in HEK293, U2OS or fibroblasts cells for 3 days if not indicated otherwise. Bars indicate the mean of measurements with cells or independent experiments as dots. Statistical significance was performed using 1- or 2-way ANOVA, n.s. non-significant, $*p \le 0.05$, $**p \le 0.01$, $***p \le 0.001$, $****p \le 0.0001$.

glucose uptake and rate of glycolysis, cytosolic pH alkalinization might promote this phenomenon as glycolytic enzymes generally perform most efficiently in a slightly alkaline environment (around pH 7.5–8.0)[68,69]. A partial lysosomal defect, therefore, might present a pre-adaptive state to sustain mitochondrial insults. Furthermore, this altered metabolism limits mitochondrial reliance and suppresses stress pathways commonly upregulated—and potentially futile— upon mitochondrial dysfunction[70–74].

Whilst we demonstrate the beneficial effect of partial v-ATPase inhibition on multiple cell types with mitochondrial translation defects, the validity of our results currently is limited to mitochondrial translation defects in cultured cells, necessitating further validation in organisms. The increase in $\Delta\Psi_m$ by low v-ATPase deficiency might also decrease vulnerability to other mitochondrial dysfunctions, as ATP6AP1[KO] cells are insensitive to artificially lowering $\Delta\Psi_m$ by FCCP, providing a link to $\Delta\Psi_m$-dependent OPA1 processing rescuing organelle architecture[75]. Furthermore, mitochondrial membrane potential has a direct role in the quality control of mitochondrial protein biogenesis[17,76].

The cost of sustained partial v-ATPase inhibition on overall growth and suppression of anabolic pathways could be mitigated and remain pathophysiologically low in postmitotic tissues of mitochondrial dysfunction. The picomolar potency required for low, sustained inhibition, combined with the availability of FDA-approved v-ATPase inhibitors, highlights potential therapeutic applicability.

Collectively, our findings reveal a mechanistic link between v-ATPase function, $\Delta\Psi_m$ and cristae architecture. We identify the v-ATPase as an extra-organellar modulator that functions as a cellular rheostat, shifting metabolic fluxes toward a low-anabolic, stress-resistant state in response to mitochondrial dysfunction-induced pathways, reshaping metabolism and enhancing cell fitness. Notably, we demonstrate that partial inhibition of v-ATPase confers metabolic resilience across diverse models of mitochondrial dysfunction, including primary models of mitochondrial translation defects. These findings have significant therapeutic implications, emphasizing the need for nuanced strategies when targeting the v-ATPase in cancer and in conditions where mitochondrial dysfunction is a critical determinant of disease progression and treatment response—such as lysosomal storage disorders—underscoring the need to elucidate the interplay between v-ATPase activity and mitochondrial function.

## Methods
### Cell lines and maintenance
Human embryonic kidney (HEK293) and human osteosarcoma (U2OS) cells were obtained from ATCC, STR authenticated and tested for mycoplasma contamination. Healthy- and patient-derived fibroblasts were obtained as indicated in the respective studies and tested for mycoplasma contamination. Cells were maintained in DMEM supplemented with 10% fetal bovine serum, sodium pyruvate, glutamine, non-essential amino acids, and uridine. Cells were maintained in a humidified incubator with 5% $CO_2$ at 37 °C. Cells were cultured in high glucose (25 mM) if not mentioned otherwise, supplemented with pyruvate to allow for NADH to $NAD^+$ recycling and pyruvate-dependent aspartate biosynthesis[77,78].

### Determination of optimal drug concentrations for CRISPR screen
HEK293 cells were seeded in 6-well plates at a density optimized to remain below 90% confluency after 72 h of treatment with minimal concentrations of drugs either inhibiting mitochondrial translation (chloramphenicol) or inducing mitochondrial mistranslation (actinonin). Cells were trypsinized, collected, counted, and re-seeded in a new 6-well plate every 72 h for 15 days. Cell lysates were collected at day 15 and analyzed by immunoblotting. Growth curves for cumulative differences in growth under drug treatments for 15 days were determined by cell counting and were plotted as the cumulative population doubling over time. Population doubling was calculated as below:

$$\text{Population doubling} = (\log(\text{cell number Day 3}) - \log(\text{cell number Day 0}))/\log(2)$$

### Determination of infection conditions for CRISPR-pooled screen
The Brunello CRISPR knockout library contains an average of 4 guides per gene for a total of 77,441 sgRNAs and 1000 control guides. For optimal infection conditions of 30–50% infection efficiency (MOI > 0.3, 0.5<), $5 \times 10^6$ cells were seeded in 15 cm diameter plates and infected 24 h post-seeding with different virus volumes (0, 50, 150, 450 µL) with a final concentration of 8 µg/mL polybrene in HEK293 cells. 24 h post-infection, cell culture medium was changed to puromycin-containing medium at 2 µg/mL final concentration. Cells were counted 72 h post-selection, and the infection efficiency was determined as the cell count of survival in plates with and without puromycin selection for the plates containing the same volume of virus. The volume of virus that yielded 30–50% infection efficiency was used for screening.

### Genome-wide CRISPR screen
Screening-scale infections were carried out using the pre-determined viral volume in the same 15-cm plate format as used for viral titration. Infections were performed with $1.5 \times 10^8$ cells per replicate (in three independent replicates) to achieve $4 \times 10^7$ surviving cells for a sgRNA representation > 500× following puromycin selection. Medium was replaced 24 h post-infection with medium containing puromycin, and the puromycin-selected cells were expanded for 7 days. The resulting clonally expanded cellular pool was separated into respective conditions, each performed in three independent replicates. After selection was complete ($T_0$), $4 \times 10^7$ of HEK293 cells were seeded in 15 cm plates per condition and per replicate. Media with EtOH-treated control, 5 µg/mL chloramphenicol, 15 µM ACT and the combination of 5 µg/mL chloramphenicol and 15 µM ACT were added to the cells. Cells were passaged in fresh media containing drugs every 72 h at a density of $4 \times 10^7$ per replicate to maintain sgRNA coverage. Cells were harvested 15 days after initiation of treatment. The relative abundance of all sgRNA was ranked on Sigma plots.

### gRNA pool amplification and sequencing
Genomic DNA (gDNA) was isolated from $4 \times 10^7$ cells per condition and per replicate using the Nucleospin blood XL kit according to the manufacturer's protocol. PCR was performed using universal primers

recognizing the unique sgRNA-independent lentiviral integration sequence (refer to Supplementary Table 1). Column-purified PCR samples were sequenced using PCR-free library preparation and PE150 on an Illumina Novaseq with 30 M reads per replicate.

## Bioinformatic analysis

Sequencing reads were quality-checked (Phred score > 20), processed in a single batch, de-indexed, and adapter-trimmed. For each gene, the normalized $\log_2$ fold-change in abundance of its four sgRNAs was calculated. No filtering for lowly expressed genes in HEK293 cells was applied. Gene-level scores were obtained by averaging across triplicates and subsequently Z-score transformed[24]. Z-scores for each treatment condition were normalized and compared against the ethanol (ETOH) control. In total, 19,112 genes were recovered, with an average of 12 million mappable reads per replicate (77,441 sgRNAs—four per gene—including 1000 control sgRNAs), providing an average library coverage of ~150×.

## Specific CRISPR-Cas9 knockouts

Lentiviral particles to confirm selected sgRNA were produced from the SangerKO CRISPR library (sequences of sgRNAs are provided in the Supplementary Table 1). Stably Cas9-expressing cells were produced in-house, and the control or WT indicated in the figures are stably Cas9-expressing. Cells were selected with 2 μg/mL puromycin 24 h post-infection and selected for 72 h. Using the serial dilution method, single cells were seeded in 96-well plates, and gene disruption efficiency was verified by protein immunoblotting. The cell lines generated in this study are found in the source data file and are shared upon request.

## Drug treatment and rescue experiments

Cells were seeded in 6-well plates at a density optimized to remain below 90% confluency after 72 h of treatment. Drug concentrations were determined in wild-type (WT) cells by assessing growth rates over a 72 h exposure. For rescue experiments with bafilomycin A1 and concanamycin A, drug concentrations were determined as the lowest concentration of drug not impairing growth over 72 h treatment. Concentrations of ACT were optimized to impair to the same extent as in HEK293 treated with 15 μM ACT (loss of ≈40% growth). Population doubling was calculated as above. If not mentioned otherwise, standard concentrations of 15 μM ACT and 5 μM CAP were used.

## Immunoprecipitation

HEK293 cells were transfected with an ATP6AP1-FLAG expression vector and cultured for 96 h with or without ACT. Cells were washed twice with ice-cold 1× PBS and lysed for 30 min on ice in NP-40 buffer (50 mM Tris-HCl, pH 7.5; 150 mM NaCl; 1% NP-40) containing a protease inhibitor cocktail. Lysates were cleared by centrifugation (14,000 × g, 10 min, 4 °C) and incubated with anti-FLAG antibody for 1 h at 4 °C with gentle rotation. Protein G Sepharose 4 Fast Flow beads, prewashed three times with lysis buffer, were added and rotated for an additional 1 h at 4 °C. Beads were then washed three times with ice-cold 1× PBS, and bound proteins were eluted for subsequent mass spectrometry (MS) analysis as described below.

## Label-free quantitative proteomics

Label-free quantitative proteomics was performed as described[79]. Briefly, cells were cultivated in 6-well plates in quadruplicates for each condition, washed with 1× PBS and scraped into 1 mL of warm medium. Cells were then pelleted, washed once with ice-cold 1× PBS, before RIPA buffer was added to the cell pellet for protein extraction. For cell pellets, protein concentration was estimated using a BCA assay, and for each replicate, an equal amount of protein (10 μg) was precipitated onto amine beads as previously described[80]. The precipitated proteins were dissolved in 50 mM ammonium bicarbonate,

reduced, alkylated and digested with trypsin (1:50 enzyme: protein ratio; Promega) at 37 °C overnight. Digested peptides were acidified and desalted by homemade C18 stage tips, or the peptides were loaded into Evosep C18 tips (IP). Conditions of the screen were analysed using timsTOF fleX (Bruker) coupled to Evosep One (Evosep), the ATP6AP1[KO] dataset with timsTOF flex coupled to nanoElute (Bruker) and IP with timsTOF PRO2 (Bruker) coupled to nanoElute (Bruker). For cells, the timsTOF flex was operated in DDA-PASEF mode. Mass spectra for MS and MS/MS scans were recorded between m/z 100 and 1700. Ion mobility resolution was set to 0.60–1.60 V·s/cm over a ramp time of 100 ms. Data-dependent acquisition was performed using 10 PASEF MS/MS scans per cycle with a near 100% duty cycle. A polygon filter was applied in the m/z and ion mobility space to exclude low m/z, singly charged ions from PASEF precursor selection. An active exclusion time of 0.4 min was applied to precursors that reached 20,000 intensity units. Collisional energy was ramped stepwise as a function of ion mobility. For IP, the timsTOF Pro2 mass spectrometer was operated in DIA-PASEF mode. Mass spectra for MS were recorded between m/z 100 and 1700. Ion mobility resolution was set to 0.85–1.30 V s/cm over a ramp time of 100 ms. The MS/MS mass range was limited to m/z 475–1000, and ion mobility resolution to 0.85–1.27 V s/cm to exclude singly changed ions. The estimated cycle time was 0.95 s with 8 cylces using DIA windows of 25 Da. Collisional energy was ramped from 20 eV at 0.60 V s/cm to 59 eV at 1.60 V s/cm. Raw files from LC-MS/MS analyses for cell pellets were submitted to MaxQuant software (ver 1.6.17.0, ver 2.0.3.0) for protein identification and label-free quantification with parameters: Carbamidomethyl (C) as a fixed modification and protein N-acetylation and methionine oxidation as variable modifications with a first search error window of 20 ppm and main search error of 6 ppm. Trypsin without proline restriction cleavage option was used, and two allowed miscleavages. Minimal unique peptides were set to one, and false discovery rate (FDR) allowed 0.01 (1%) for peptide and protein identification. The Uniprot human (v.2020) database was used. Generation of reversed sequences was selected to assign the FDR rate. For IP, the raw files were submitted to DIA-NN (version 1.8.1) for protein identification and label-free quantification using the library-free function. The UniProt human database (v.2020, UniProt consortium, European Bioinformatics Institute, EMBL-EBI, UK) was used to generate a library in silico from a human FASTA file. Carbamidomethyl (C) was set as a fixed modification. Trypsin without proline restriction cleavage option was used, with one allowed miscleavage and peptide length range set to 7–30 amino acids. The mass accuracy was set to 15 ppm, and the precursor FDR allowed was 0.01 (1%). LC-MS/MS data quality evaluation and statistical analysis were done using software Perseus v.1.6.15.0

## Untargeted metabolomics

Cells were seeded in a 12-well plate with an optimized cell confluency (<90%) after treatment. Cells were washed twice with 75 mM ammonium bicarbonate (pH 7.4), and metabolome extraction was performed on a plate with 40:40:20 acetonitrile:methanol:water for 10 min at −21 °C and repeated once. Extracts were centrifuged for 2 min at 18,000 × g and the supernatant was transferred to a new Eppendorf tube and stored at −20 °C until analysis. Untargeted metabolomics was performed by flow injection-time-of-flight (TOF) MS on an Agilent 6550 QTOF instrument in negative mode[81]. Indexing of spectra was created using the Human Metabolome Database v.3.6 using a mass per charge tolerance of 0.001 m/z and isotopic correlation pattern, resulting in >1500 quantifiable peaks of putatively annotated metabolites. Quantification and metabolic pathway analysis were performed using MetaboAnalyst 5.0 or IPA v.01-22-01, respectively.

## $^{13}$C-Metabolic flux analysis

Cells were seeded in a 6-well plate with an optimized cell confluency (<90%) after treatment. Cells were washed once with 1× PBS, and

1,2-$^{13}$C$_2$ labeled glucose medium was added to the cells for 60 min. The medium was removed, and the cells were washed with ice-cold 1 x PBS before the cells were scraped with 80:20 acetonitrile:water and transfer in an Eppendorf tube. The cells were centrifuged for 10 min at 4 °C at 10,000 × $g$ and 200 µL of supernatant was transferred into a microvial and stored at −80 °C until processing. Samples were analyzed on a Thermo Q Exactive Focus Quadrupole Orbitrap mass spectrometer coupled with a Thermo Dionex UltiMate 3000 high-performance liquid chromatography (HPLC) system (Thermo Fisher Scientific). The HPLC was equipped with a hydrophilic ZIC-pHILIC column (150 × 2.1 mm, 5 µm) with a ZIC-pHILIC guard column (20 × 2.1 mm, 5 µm, Merck Sequant). A 5 µL sample was injected into the liquid chromatography-mass spectrometry (LC-MS) instrument after quality controls in randomized order. A linear solvent gradient was applied in decreasing organic solvent (80−35%, 16 min) at 0.15 mL min−1 flow rate and 45 °C column oven temperature. Mobile phases were aqueous 200 mmol per liter ammonium bicarbonate solution (pH 9.3, adjusted with 25% ammonium hydroxide), 100% acetonitrile, and 100% water. Ammonium bicarbonate solution was kept at 10% throughout the run, resulting in a steady 20 mmol per liter concentration. Metabolites were analyzed using a mass spectrometer with a heated electrospray ionization source using polarity switching and the following settings: resolution of 70,000 at m/z of 200; spray voltages of 3400 V for positive and 3000 V for negative mode; sheath gas of 28 arbitrary units (AU) and auxiliary gas of 8 AU; vaporizer temperature of 280 °C; and ion transfer tube temperature of 300 °C. The instrument was controlled using Xcalibur 4.1.31.9 software (Thermo Scientific). Metabolite peaks were confirmed using commercial standards (Sigma-Aldrich). Peak integration and metabolite isotopologue identification were accomplished using TraceFinder 4.1 SP2 software (Thermo Scientific). Specificity of labeled peaks and isotopologues was confirmed using cell line and blank control samples. Data processing was performed in R version 4.5.1 with the tidyverse 2.0.0 package family. Natural $^{13}$C abundance and tracer impurity were addressed with correction calculations using the IsoCorrectoR R-package[82] and normalized to total metabolite intensity. One sextuplicate from each condition was discarded due to low metabolite yield from the pressure increase. Data are represented as the respective istopologue ratio indicated or as the total sum of all masses.

## MtDNA copy number quantification
MtDNA was quantified using qPCR as described[83]. Briefly, genomic DNA was extracted using the DNeasy blood and tissue kit according to the manufacturer's instructions. Genomic DNA was diluted to 1 ng/µL concentration, and 8 µL were used per qPCR reaction, including 10 µL of SYBR green and 2 µL of 10 µM primer mix (sequence provided in Star methods). qPCR was performed using a CFX96 Biorad thermal cycler. The PCR program used for amplification consisted of 7 min at 95 °C, followed by 39 cycles of 10 s at 95 °C, 30 s at 60 °C. The run was completed with 10 s at 95 °C followed by 5 s at 65 °C and 0.5 °C temperature increment to 95 °C.

## High-resolution oxymetry
Cellular oxygen consumption rates were measured using an Oroboros O2k instrument as described[84]. In brief, harvested cells were resuspended in respiration medium composed of 0.5 mM EGTA, 3 mM magnesium chloride, 60 mM potassium lactobionate, 20 mM taurine, 10 mM potassium dihydrogen phosphate, 20 mM HEPES, 110 mM sucrose, and 1 g/L fatty acid−free BSA (pH 7.1). Routine respiration was first assessed, after which oligomycin (10 nM) was added to determine leak respiration. Maximal uncoupled respiration was then evaluated by stepwise titration of carbonylcyanide p-trifluoromethoxyphenylhydrazone (FCCP; 0.5 µM increments). Residual respiration was quantified following the addition of rotenone (0.5 µM) and antimycin A (2.5 µM). Finally, complex IV activity was determined by adding ascorbate (2 mM) and N,N,N′,

N′-tetramethyl-p-phenylenediamine dihydrochloride (TMPD; 0.5 mM), followed by azide (10 mM).

## Immunoblotting
Protein extracts were isolated from cells with RIPA buffer in the presence of protease and phosphatase inhibitors. Protein concentrations were determined using a BCA assay. Protein extracts were separated by SDS-PAGE and transferred to a PVDF membrane using the trans-blot turbo transfer system. Equal loading was determined by stain-free protein intensities. Following blocking in TBS-T containing 5% skim milk for 1 h at RT, the membranes were probed with primary antibodies overnight at 4 °C and secondary antibody (goat anti-rabbit or goat anti-mouse IgG HRP) for 1 h at RT. Detection was done using Clarity Western ECL Substrate with the ChemiDoc XRS+ System. Uncropped immunoblots are provided in the Source data file. Refer to the Supplementary Table 1 for all antibodies used in this study.

## Blue-native acrylamide electrophoresis
For native respiratory complex analysis[85], cell pellets from one 10 cm$^2$ dish per replicate were placed on ice and swelling induced by adding an equal volume of hypotonic RSB Hypo Buffer (10 mM NaCl, 1.5 mM MgCl$_2$, 10 mM Tris-HCl, pH 7.5) for 5-10 min. Next, the cells were homogenized manually by ten strokes with a 2 mL Teflon potter−Elvehjem. Following homogenization, 2.5× MS homogenization buffer (525 mM mannitol, 175 mM sucrose, 2.5 mM EDTA, 2.5 mM DTT, 12.5 mM Tris-HCl, pH 7.5) was added for a final concentration of 1× MS homogenization buffer. The homogenate was then centrifuged at 1300 × $g$ for 3 min at 4 °C. The supernatant, containing mitochondria, was collected and further centrifuged at the same speed to pellet any remaining heavy contaminants. The supernatant was transferred to a new tube, and the mitochondria were pelleted by centrifugation at 15,000 × $g$ for 2 min at 4 °C. The pellets were washed in 1× MS buffer and centrifuged again. Mitochondria were resuspended in aminocaproic acid buffer (1.5 M aminocaproic acid, 50 mM Bis-Tris, pH 7.0) containing protease inhibitors. Protein concentration was quantified using the BCA assay and aliquots of 250 µg protein were solubilized with digitonin (1:6 digitonin to protein ratio) for 10 min on ice. Post-incubation, the samples were centrifuged at 20,000 × $g$ for 30 min at 4 °C to clarify the extracts. Samples were mixed with 10× native loading buffer (1.5 M Aminocaproic acid, 50 mM Bis-Tris, 0.5 mM, EDTA, 5% (w/v) Serva Blue G, pH 7.0) and electrophoresed on 3−10% self-cast gels. After the run gels were semi-dry blotted onto PVDF membrane. The membranes were blocked in TBS-T with 5% milk powder for 1 h, incubated in primary antibody overnight at 4 °C and detected with the respective HRP-secondaries using chemiluminescence.

## Mitochondrial membrane potential (ΔΨ$_m$) determination and acidic organelle tracking
Cells were seeded in a 24-well plate and incubated with 100 nM TMRM or LysoTracker Red DND-99 or 33 µM of acridine orange for 30 min at 37 °C in phenol-free medium (and addition of Hoechst). Cells were washed once with warm phenol red-free medium prior to imaging with an Evos FL Digital microscope or Incucyte S3 with 20× objectives. A minimum of 100 cells per group were analyzed using relative signal intensities or analyzed with the Incucyte 2022B software as the integrated intensity normalized to the phase area. For structural assessment, LysoTracker Red DND-99 was incubated at 200 nM for 1 h, fixed for 10 min in 4% PFA and directly imaged in a Zeiss Axiovert.

## Assessing lysosomal and cytosolic pH
LAMP1-localized vesicle pH was ratiometrically determined with a stably expressed construct containing lumen-oriented mTFP1 and cytosolic mCherry (pLJM1-FIRE-pHLy). To estimate the cytosolic pH, cells were incubated with 2.5 µM of BCECF AM for 30 min in 37 °C and then analyzed in Live cell imaging solution with the Multimode plate

reader EnSpire (PerkinElmer), and the ratio was calculated from dual-excitation with $\lambda_1 = 490$ nm and $\lambda_2 = 440$ nm and fixed emission at 535 nm. Standard curves used to estimate both lysosomal and cytosolic pH were created by a similar analysis of cells incubated with a series of pH calibration buffers supplemented with 10 µM valinomycin and 10 µM nigericin (Intracellular pH calibration kit).

### Electron microscopy
Cells were seeded on thickness #1 coverslips, treated for the indicated time and fixed in cacodylate buffer containing 2% glutaraldehyde for 25 min at RT, washed and stored until processing of the samples. Routine Epon embedding was performed, and standard 60 nm thin sections were obtained on an ultramicrotome. Ultrathin sections were imaged on a Jeol JEM-1400 at 100 kEV equipped with a Gatan Orius SC 1000B CCD-camera. Micrographs were post-processed using iTEM and brightness-adjusted using Adobe Photoshop 2023 software.

### Immunofluorescence microscopy
Preparation of immunofluorescence samples was performed as described[79]. Briefly, cells were cultured on coverslips and fixed with 4% paraformaldehyde for 10 min at RT and subsequently washed three times with 1×PBS. The cells were permeabilized and blocked with 0.1% Triton-X-100 and 10% horse serum for 1 h, washed with 1% horse serum, 0.1% Triton-X-100, 1% BSA in 1×PBS for 5 min. Incubation with primary antibodies was performed overnight at 4 °C with 1×PBS containing 1% BSA, 1% horse serum, 0.1% Triton-X-100 in 1×PBS, then washed three times with 1×PBS containing 1% BSA for 5 min and incubated with secondary antibodies for 1 h in 1×PBS containing 1% BSA. The cells were then washed twice with 1×PBS for 5 min, counterstained with Hoechst (1:5000) in 1×PBS for 5 min, washed twice with 1×PBS for 5 min and mounted in VectaShield. Slides were imaged with Zeiss Axio Observer Z1 fluorescence microscope, oil objective 63×. Mitochondrial fragmentation and co-localization analyses were performed with a minimum of 100 cells per group using CellProfiler 4.2.5 (analysis pipelines available on request). Analysis of other quantitative measures were performed from maximal projections of z-stacks using FIJI v.2.14.0.

### Digitonin fractionation
Cells were lysed for 30 min on ice with extraction buffer (250 mM sucrose; 20 mM HEPES; 10 mM KCL; 1.5 mM MgCl$_2$; 1 mM EDTA; 1 mM EGTA, protease inhibitors) containing the respective concentration of digitonin, followed by centrifugation for 10 min at 12,000×$g$. Supernatants were transferred into a new tube containing Laemmli buffer and 2-mercaptoethanol. Samples were separated by SDS-PAGE for immunoblotting.

### Statistics and reproducibility
All graphical representations were performed in Adobe Illustrator 29.3.1, with statistic and data representations performed in Prism v10.4.2, if not indicated otherwise. Structural representations were created using Pymol v.2.4.1. with indicated PDB entries. All data representations are shown as mean values ± standard deviation (SD), calculated using GraphPad Prism v10.4.2 with $p$ value < 0.05 considered as threshold for statistical significance. $P$ values are provided within each figure legend, together with the statistical test performed for each experiment: two tailed one-sample, unpaired two-tailed Student's t-test or two-way ANOVA followed by post hoc test for multiple comparisons calculated in GraphPad Prism v10.4.2, if not indicated otherwise. All replicate values represented as dots indicate independent experiments with bar plots indicating mean ± SD, if not mentioned otherwise in the figure legend. Derived statistics are based on averaged values across biological replicates rather than pooled technical replicates. For differential intensity analyses, proteomic and metabolomic data were log-transformed and quantile-normalized. The resulting data were processed through a standard limma-based pipeline using the edgeR package. Linear model fitting was applied to each protein and metabolite using generalized least squares according to the experimental design. When both HEK293 and U2OS cell lines are included, we incorporated a covariate into the design matrix to account for the origin of the cell lines. We defined contrasts for the empirical Bayes procedure to obtain moderated t-statistics and Benjamini–Hochberg adjusted $p$ values. Pathway-level significance was assessed using the EGSEA R package. For metabolite data, we use over-representation analysis. No statistical method was used to predetermine sample size. The investigators were not blinded to allocation during experiments and outcome assessment.

**Ethical approval.** All collaborators of this study have fulfilled the criteria for authorship required by Nature Portfolio journals have been included as authors, as their participation was essential for the design and implementation of the study. Roles and responsibilities were agreed among collaborators ahead of the research. This research was not severely restricted or prohibited in the setting of the researchers, and does not result in stigmatization, incrimination, discrimination or personal risk to participants. Local and regional research relevant to our study was taken into account in citations.

### Reporting summary
Further information on research design is available in the Nature Portfolio Reporting Summary linked to this article.

## Data availability
All data supporting the findings of this study, including processed results and figure source data, are provided in the Supplementary Information files. CRISPR screening raw sequencing data generated in this study has been deposited in the NCBI Gene Expression Omnibus (GEO) under accession number GSE261899. Proteomic raw data sets have been deposited in PRIDE under accession numbers PXD057148 and PXD050818 without access restrictions. Source data are provided with this paper. Further information and requests for resources and reagents should be directed to the Lead contact Christopher Jackson (christopher.jackson@helsinki.fi). Source data are provided with this paper.

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

## Acknowledgements

The authors wish to thank Dr. Brendan Battersby for constructive discussions and Tarja Grundström for technical assistance throughout the project. This work was supported by funding from the Academy of Finland (to C.B.J., decision #336455), the Magnus Ehrnroot Foundation (to C.B.J.), the Jane and Aatos Erkko Foundation (to C.B.J., decision #230004), the Research Council Finland (to C.B.J., decision #368527; to V.H., decision #312439) and a Doctoral fellowship of the Integrated Life Sciences Graduate School (to R.A.). We acknowledge the Turku Microscopy Centre and the Electron Microscopy Unit of the Institute of Biotechnology (EMBI), University of Helsinki, for technical support and access to laboratory facilities. The Human Brunello CRISPR knockout pooled library was a kind gift from David Root and John Doench. Individual sgRNAs were from the Sanger gRNA library, distributed by Genome Biology Unit core facility supported by HiLIFE and the Faculty of Medicine, University of Helsinki, and Biocenter Finland. Open access funding was provided by Helsinki University Library. MS-based proteomic analyses were performed by the Proteomics Core Facility, Department of Immunology, University of Oslo/Oslo University Hospital, which is supported by the Core Facilities program of the South-Eastern Norway Regional Health Authority. This core facility is also a member of the National Network of Advanced Proteomics Infrastructure (NAPI), which is funded by the Research Council of Norway INFRASTRUKTUR-program (project number: 295910).

## Author contributions

M.G.—conceptualization, methodology, analysis; R.A., K.D., A.M., S.S.—analysis; N.Z., T.A.N., S.K.S., M.K., J.J., H.V., C.J.C.—analysis, methodology; C.B.J.— conceptualization, methodology, analysis, supervision, resources.

## Competing interests

The authors declare no competing interests.

## Consent to participate

Consent has been obtained previously for the use in patient-derived fibroblasts for research purposes.
