## [Transparent Peer Review file · Nature Communications]

Vacuolar-type H⁺-ATPase-mediated extra-organellar buffering resolves mitochondrial dysfunction

Corresponding Author: Dr Christopher Jackson

Version 0:

Reviewer comments:

Reviewer #1

(Remarks to the Author)
Comments to the authors

In this manuscript, through CRISPR/Cas9 screen, the authors identify that the inhibition of vacuolar v-ATPase can restore the cell fitness under mitochondrial translation inhibitors-treated conditions. The study contains several interesting findings. However, this reviewer feels the current version of the manuscript is not easy to follow everything for several reasons. Also, some unclear points were observed. Please see specific comments below.

Major points

1. Each figure panel is too small to read. For example, Fig. S5 contains so many information, and thus it is very hard to capture the main argument of the figure. I would suggest the authors to make each figure more understandable.
2. The authors should focus on simple but sufficient description as much as possible to avoid misunderstanding of readers. For example, "CAP had no added effect on growth in ACT conditions but altered overall protein expression and lead to partial rescue of mitochondrial fragmentation with near to complete loss of translation at 5 µg/mL (Figure 1D; S1A, B)" on page 6. Several observations are described in one sentence. I think "mitochondrial fragmentation" is not properly addressed in indicated figures, although I guess the authors mention the OPA1 band pattern as an indicator of mitochondrial morphology. It is better to explain what kinds of protein expression are altered. Also on page 6, following sentence needs more simplified understandable description: "All drug concentrations were adopted to comparable growth inhibition to avoid subsequent proliferation-induced hit biases - prominent with loss of mitochondrial genes -, whilst ensuring sufficient resolution of the screen at low dose (Figure 1D)". Please notice that these are just part of examples.
3. The purpose and the interpretation of the series of experiments in Fig. S3K-O are not clear.
4. From several analyses in Fig. 4 and Fig. S5, the authors concluded "v-ATPase-dependent deficiency remodels cellular metabolism beneficial under mitochondrial dysfunction". But it is not clear what kinds of beneficial metabolic changes are actually induced, and how those are linked with the observed restored mitochondrial and cellular fitness. In Abstract, the authors mentioned that "v-ATPase regulates the cellular redox environment", but is this proved in this figure? If not, where is the evidence?
5. OPA1 is known to be cleaved by OMA1 in response to mitochondrial depolarization. In Fig. S6A, this does not seem to be induced even in WT cells.
6. In Fig. S6B, the authors mentioned that v-ATPase inhibition increases OPA1 expression. But in Fig. 3F, the authors argued that ATP6AP1 deletion (kind of v-ATPase inhibition) prevents the OPA1 processing. Which is the authors' conclusion?
7. In Fig. 5E and 5F, the authors focused on extracellular pH. But the underlying logic that leads to this experiment is not explained clearly.
8. In Fig. S6G-I, the authors showed that ATP6AP1 KO cells are also resistant to staurosporine or FCCP-treated conditions. Does this mean the beneficial effects of v-ATPase inhibition are not specific to mitochondrial dysfunction that is caused by mitochondrial translation inhibitors?
9. Higher mitochondrial membrane potential in ATP6AP1 KO cells is very clearly presented (Fig. 5A). This is a key finding of this study. However, the underlying molecular mechanisms are not described well in the current study.

Minor points

1. Same immunoblot panels seem to be used in Fig. 1B and Fig. 2A. At least, the authors should mention this in the figure legend, if that is the case.

2. Fig. S3J is not mentioned in the text.

3. Several panels in Fig. S6 are not properly cited in the text: panels E and F are not mentioned, panels O and P do not exist.

Reviewer #2

(Remarks to the Author)

In this manuscript, the authors seek to understand the cellular consequences of mistranslation of mitochondrial proteins through a genome-wide CRISPR screen in the presence of actinonin, an inducer of mistranslation, with and without the mitochondrial translation inhibitor chloramphenicol. In addition to several expected hits, guides against several V-ATPase subunits emerged as potential suppressors of actinonin effects on mitochondria. The authors choose to focus on effects of ATP6AP1, also known as Ac45, a V-ATPase assembly factor that was shown to be part of the lysosomal V-ATPase in recent structures. Extensive characterization of mitochondrial structure, mitochondrial proteins such as Opa1, and the overall metabolic state of cells in wild-type and ATP6AP1 knockout cells are provided. Unfortunately, many of these experiments are either not tied back to mechanism or provide few insights into the mechanistic consequences of mistranslation and how reduced V-ATPase activity might compensate for the effects of actinonin. In their final analysis, they conclude that the ATP6AP1 knockout or low concentrations of the inhibitors concanamycin and bafilomycin may alter overall pH homeostasis, lowering cytosolic pH and therefore increasing mitochondrial membrane potential. This is a reasonable hypothesis for a potential effect from V-ATPase activity, but it is not conclusively tested here. As a result, the authors have a potentially interesting result from their screen, but it provides no real mechanistic insight.

The following points should be noted if the paper is revised.

1. The authors report that their ATP6AP1 knockout is hypersensitive to V-ATPase inhibitors relative to wild-type cells (lines 290-292). This result is disturbing, as it indicates that there must still be residual V-ATPase activity somewhere in the cell, since these inhibitors are highly specific. This key point is not addressed. It is possible that the knockout only affects a subpopulation of V-ATPases that require ATP6AP1 or else the ATP6AP1 knockout is not complete. Either possibility complicates interpretation of most of the subsequent results.

2. In Figure 4G and H, the authors show that the ATP6AP1 knockout still accumulates LysoTracker in acidic compartments, and may in fact accumulate more. This is inconsistent with the expected loss of vacuolar acidification in this mutant. The presence of other core V-ATPase subunits in the original CRISPR screen would suggest that loss of V-ATPase activity and acidification is what suppresses effects of actinonin.

3. The possibility that perturbed pH homeostasis might be responsible for the apparent suppression of actinonin effects is reasonable, but is not tested thoroughly. Both lysosomal pH and cytosolic pH could be determined quantitatively with ratiometric pH sensors. However, even labels like pHrodo-green that are not ratiometric can be calibrated to pH. Simply reporting mean fluorescence intensity as in Figure 5H is not adequate.

4. There is no question that the authors did a lot of work for this paper, but the Figure legends could be improved to make it easier to determine exactly what is being shown.

Reviewer #3

(Remarks to the Author)

Using a genome-wide CRISPR screen, the author identified H⁺-type vacuolar (v-)ATPase subunits as regulators of mitochondrial dysfunction by disrupting the inner mitochondrial membrane. The author elucidated the mechanism by which v-ATPase rescues mitochondrial dysfunction and introduced the concept that lysosome function can regulate mitochondrial membrane potential. This work comprises a substantial amount of data and is interesting in terms of inter-organellar communication. However, the following issues need to be addressed before publication.

Major Comments

1. PPP Pathway Activation Evidence: In Figures 4A-C, the author claims that the pentose phosphate pathway (PPP) is activated in actinonin (ACT)-treated v-ATPase-deficient cells due to a decrease in total amino acid concentration. If the PPP is activated, the NADP/NADPH ratio should decrease, and the GSH/GSSH ratio should increase. However, in Figure 4B, the GSH/GSSH ratio decreases in ACT-treated v-ATPase-deficient cells compared to non-treated v-ATPase-deficient cells. To substantiate the claim of PPP activation, the NADP/NADPH ratio should be measured in addition to GSH/GSSH.

Additionally, performing a label-free proteome analysis to examine changes in protein levels involved in glycolysis (e.g., ALDOA, PKM2, LDHA) and the PPP (e.g., G6PD, PGD) under each cell condition would support this hypothesis.

2. Sample Consistency in Fig. 4C: Clarification is needed regarding the sample differences between "WT ACT" in the upper panel and "ACT" in the lower panel of Fig. 4C. If both data sets represent normal HEK293 cells treated with ACT, the observed differences require an explanation or rectification.

3. LAMP2 Expression in Fig. 4F: The claim of increased LAMP2 expression in ATP6AP1KO cells in Fig. 4F appears unclear. This should be substantiated by also presenting LAMP2 levels from the label-free analysis in Figure 4D for better clarity and support.

4. LysoTracker Signal Interpretation in Fig. 4G and 4H: Figures 4G and H show an increased LysoTracker signal, indicating more acidic vesicles. Since LysoTracker signal intensity is pH-dependent, this could result from either an increase in the number/volume of acidic vesicles or a decrease in lysosomal pH, or both. Analyzing the number and volume of lysosomes per cell in Fig. 4G and discussing the metabolic implications of these changes would provide a more comprehensive

understanding.

5. LAMP1 and ATP6AP1 Co-Migration in Figure S4G: The co-migration of LAMP1 and ATP6AP1 in subcellular fractionation (Figure S4G) suggests potential interaction. If an interaction is assumed, the authors should discuss its significance and potential functional implications.

6. Intracellular pH and Mitochondrial Membrane Potential: The authors suggest that ACT treatment decreases membrane potential, and v-ATPase perturbations increase cellular pH, thus recovering mitochondrial membrane potential. Testing intracellular pH changes in ACT-treated cells using TMRM and examining intracellular pH under the conditions shown in Fig. 5B (where mitochondrial membrane potential increases in ATP6AP1KO cells treated with ACT) would validate this hypothesis. An explanation for the increased mitochondrial membrane potential under these conditions is also required.

7. pH Buffering and Cell Fitness in ACT-Treated Cells: To show the importance of pH buffering for cell fitness in ACT-treated cells, the authors should modify pH by adding HCl or NaOH to the medium and assess cell fitness. Additionally, examining other parameters such as OPA1 processing, changes in protein levels, mitochondrial fragmentation, and rerouting of glycolysis in v-ATPase-deficient cells would provide further insights.

Minor Comments

1. Western Blotting in Fig. 1B and Fig. 2A: If the same western blotting results are used in Fig. 1B and Fig. 2A, this should be explicitly stated.

2. Character Size in Fig. 2F and G: The character size in Fig. 2F and G is too small to read, and the information is not informative enough. These figures should be omitted or revised for clarity.

3. Label-Free Quantitative Proteomics in Fig. 3E: Clarify if label-free quantitative proteomics detects ATP6AP1/2 even in ATP6AP1KO conditions as shown in Fig. 2C. Also, clarify the Y-axis value and what was normalized and set to 1.0.

4. Figure Legend Discrepancy: On page 13, line 308, the text describes "72 h ACT-treated WT cells" while Figure S3B indicates "15 days treatment." Verify and correct this discrepancy. Additionally, ensure that references to Figure 3E and Figure S3B are accurate.

5. Missing Labels in Fig. 4C: The label "ACT" is missing from the right bottom, right side of Fig. 4C. This should be added.

6. Unexplained Figures S3J and S4B: Figures S3J and S4B are not explained in the manuscript. These should either be explained or removed.

7. Typographical Error in Fig. 5G: Correct the typo "Phrodo" to "pHrodo" in Fig. 5G.

8. Figure Reference Correction: On page 22, line 505, correct the figure reference from (Figure S6J-P) to (Figure S6J-N).

9. Terminology Correction: On page 23, line 526, change "inter-organelle" to "extra-organelle."

Reviewer #4

(Remarks to the Author)

Reviewer #5

(Remarks to the Author)

In this manuscript, entitled 'v-ATPase-mediated extra-organellar buffering resolves mitochondrial dysfunctions', the authors propose a new signaling mechanism where mitochondrial translation-induced stress is alleviated by the inhibition of the vacuolar (v-)ATPase, a complex responsible for the acidification of intracellular compartments. Methodologically, the authors first apply a genome-wide CRISPR screen in the presence of mitochondrial ribosome inhibitors to identify pathways/modulators of cellular fitness upon which the v-ATPase complexes emerged as an interesting candidate.

Following a wide array of cell biological and biochemical characterizations, including in knockout lines, the authors describe v-ATPase as a regulator of mitochondrial cristae morphology and mitochondrial membrane potential and suggest v-ATPase inhibition to be beneficial in specific conditions of mitochondrial dysfunction, with broader implications, including to cancer and aging.

Generally, identifying mechanisms of mitochondrial-nuclear crosstalk and adaptation to cellular stress responses is of broad interest given the breadth of conditions, where mitochondrial dysfunction has been described, including in primary mitochondrial disorders. Overall, the experiments appear to have been thoroughly done and this reviewer did not identify any obvious shortcomings concerning their execution. However, the following comments should be strongly considered to improve the manuscript.

Major comments

1. Clarity - The manuscript is very densely written and has an extremely high degree of complexity, which will make it challenging for a general audience to appreciate. This reviewer would suggest revising the text for clarity to better follow the author's reasoning and decision-making of their experimental work-up.

2. For most of the manuscript, the authors conduct experiments in 293 cells that are certainly adequate for the screen and subsequent biochemical characterizations. Cancer-derived U2OS cells are then used as a form of validation, although the scope of related experiments is relatively quite limited. In addition, the cell lines appear to be mostly conducted in the presence of high glucose, raising the overall question of how generalizable the described findings are considering the framing surrounding primary mitochondrial disorders, neurodegenerative diseases, and cancer. Specifically, can the authors confirm the buffering effect of v-ATPase-mediated in a primary cell model, including with more physiologic media conditions

(e.g., human plasma-like media)?

3. Along these lines, mitochondrial dysfunction *in vivo* may arise in response to a variety of stressors, genetic mutations in nuclear as well as mtDNA affecting protein-coding genes as well as tRNAs, hypoxia, nutrient deficiency, etc. As of the current state, it remains unclear to what extent the described pathway is of broader relevance, which is something the authors also partially elaborate on in their discussion. The author is not necessarily asking for additional experiments (which of course would also be appreciated), but the limitations in terms of the interpretation of their results should be stated very clearly.

Reviewer #6

(Remarks to the Author)

Version 1:

Reviewer comments:

Reviewer #1

(Remarks to the Author)

Comments to the authors

In the latest version of the manuscript, the authors have made substantial edits on text and figures. The content is now much improved to follow. The study is extremely dense and comprehensive. However, this reviewer feels that it lacks necessary experiments to support their main argument. I also feel that it is sometimes difficult to accept the authors' interpretation of data. I fully agree that mild inhibition of lysosomal V-ATPase function has beneficial effects on cell fitness when cells are encountered with mitochondrial stress. However, some keywords such as "organelle communication" and "retrograde signaling" may be somewhat overstatements, since several key phenotypes in ATP6AP1 KO cells identified here (for example, higher mitochondrial membrane potential indicated in Fig. 2G) are steady-state phenotypes, and the mitochondrial stress-dependent active crosstalk between the two compartments was not identified in this study. In summary, the study contains a lot of information and potentially important findings, but the authors are encouraged to reshape the overall content to support their main argument with solid data. Please see specific comments below.

Major points

1. In Fig. 2F, the authors monitored OPA1 processing as an indicator of mitochondrial morphology (line 263-264). However, as shown in Fig. S3F, tubular mitochondria are significantly reduced in ATP6AP1 KO cells from the steady state, which means the OPA1 processing is not a good indicator of mitochondrial fragmentation in this context. Therefore, the authors' interpretation of these data (line 281-283) and the model illustration (Fig. 2I) are difficult to accept.
2. In Fig. 2G, the authors indicated that ATP6AP1 KO cells have a higher mitochondrial membrane potential already at the steady state. In Fig. 3H, the authors indicated that actinonin induces cytosolic acidification upon treatment in WT cells, but not in ATP6AP1 KO cells, which is an interesting observation. These two factors (mitochondrial membrane potential and intracellular pH) are important phenotypes observed in ATP6AP1 KO cells, and the authors try to relate them to the observed mitochondrial stress resistance in ATP6AP1 KO cells. However, several uncertain points remain in this argument. 1) Are these phenotypes related to each other? If so, what are the underlying mechanisms? – This point is related to my previous comment 9. 2) There is no direct evidence that proves these two factors indeed confer the stress resistance to ATP6AP1 KO cells. Regarding this point, as I indicated in my previous comment 7, I still don't understand the purpose of the experiments conducted in Fig. S5 D-G. To test that the cytosolic pH does indeed affect the mitochondrial stress resistance of ATP6AP1 KO cells, the experiments should be performed under actinonin treatment. For example, if the cytosolic pH is adjusted to acidic in ATP6AP1 KO cells, do these cells lose their resistance to actinonin?
3. Based on the data set in Fig. S4 (and their response to my previous comment 8), the authors considered that ATP6AP1 KO cells are resistant to a wide range of mitochondrial stresses that induce mitochondrial membrane potential loss. Although the study was originally initiated by mitochondrial translation inhibitors, it is better to include these observations in the graphical abstract.

Reviewer #2

(Remarks to the Author)

In this revised manuscript, the authors have strengthened their evidence for a reciprocal relationship between mitochondrial membrane potential and morphology and lysosomal v-ATPase activity. The data are stronger, but some points in the text and figures need to be clarified. These changes should not require additional experiments but are important for the overall message of the manuscript.

1. In Figure 3F, the authors compare calibrated measurements of lysosomal pH in normal and ATP6AP1 KO cells. These measurements show that lysosomal pH increases rather modestly, by 0.6 pH units in the untreated mutant cells and 0.3 pH units in the ACT-treated cells. This relatively small increase is consistent with the ATP6AP1 KO not completely abolishing v-ATPase activity in the lysosome, as acknowledged by the authors, but the Figure 3 legend states "its loss causes organelle

alkalinization” and the authors state on line 392 that “the ATP6AP1 KO cells had an alkaline pH”. A pH of 4.9 is not alkaline. Similarly on line 299 and 302 the authors cite “loss of v-ATPase”. A strength of the paper is the determination that low concentrations of v-ATPase inhibitors, which do not abolish activity, mimic the effects of the ATP6AP1 mutant, but the statements above confuse this comparison. It would be better to refer to reduced v-ATPase activity or partial loss of lysosomal acidification by ATP6AP1 KO throughout the paper.

2. Figure 3H shows that WT cells experience cytosolic acidification upon ACT treatment, while ATP6AP1 KO does not. This is an important result that deserves a little more discussion. It could indicate an issue with cellular buffering capacity as the authors suggest, but on the simplest level, a lower cytosolic pH (resulting to an increased Δ pH across the mitochondrial membrane) might actually increase, not decrease, mitochondrial membrane potential, so clarification is needed.

Minor points:

1. Figure S3A and lines 258-260—azide is not a V-ATPase inhibitor in cells. In reference 29, it was only shown to inhibit proton pumping of the reconstituted clathrin-coated vesicle ATPase. I would omit this inhibitor.
2. The legends of Figure 4C-H do not match the figure.
3. Lines 560-562 and Figure 5E and F—it doesn't look like there is any significant change in mitochondrial membrane potential in the U2OS cells (unless there is something missing in the Figure 5E) and the difference is very modest in Figure 5F. It might be good to soften the description of these data.
4. Lines 671-672—the “NAD⁺-mediated acidification” is used somewhat out of context here relative to the original work. This could be confusing.
5. Lines 679-680—the mutant's resistance to EtBr is consistent with its ability to maintain mitochondrial membrane potential, but does not really provide any additional information about ATP synthase or signaling pathways.

Reviewer #3

(Remarks to the Author)

We appreciate the authors' substantial revisions and efforts to improve the manuscript. However, several of our concerns remain insufficiently addressed. Additionally, the authors did not clearly indicate where changes were made in the revised manuscript, which complicates the review process. For clarity and efficiency, specific changes should be referenced ideally in the rebuttal letter (e.g., “We have added a new explanation on page XXX in the revised manuscript”). I believe that the rebuttal letter serves as a crucial platform for constructive communication with reviewers. Providing clear, thorough, and well-reasoned responses not only addresses the reviewers' concerns effectively but also contributes significantly to improving the quality and clarity of the manuscript.

While we acknowledge the authors' completion of several experiments suggested by other reviewers, the overall flow of the manuscript has become more difficult to follow. Strengthening the logical connections between results would improve the clarity and coherence of the narrative. Should the authors revise the manuscript again, we recommend addressing the following points:

PPP Pathway Activation Evidence: The authors' response—“Our ratios from untargeted metabolomics are a crude approximation and reflect a steady-state and no flux”—remains unclear for this reviewer. This explanation requires clarification. Also, I suggested here that the NADP/NADPH ratio should be considered, but the authors did not perform that experiment directly. Although the authors were not obliged to conduct every experiment suggested by reviewers, they should address the reviewer's concerns by explanation. Moreover, the authors should indicate where and how they have changed the manuscript.

Sample Consistency in Figure 4C: The authors attributed the removal of certain data to a “typo” without providing an adequate explanation. They should explain what specific errors occurred and justify the data removal, as this is crucial for transparency in the review process.

LAMP2 Expression in Figure 4F: The authors stated that they “changed the text to emphasize this” but failed to specify how or where these changes were made. A clear indication of these modifications within the manuscript is necessary.

Lysotracker Signal Interpretation in Figures 4G and 4H: Although new experiments were performed, my question is “either an increase in the number/volume of acidic vesicles or a decrease in lysosomal pH, or both” “discussing the metabolic implications of these changes”. Please explain this point.

Intracellular pH and Mitochondrial Membrane Potential: The authors employed a method to quantify lysosomal pH and intracellular pH, allowing more detailed measurement of pH changes. While this is a great attempt, it raises a new question. In a previous manuscript, the authors suggested that v-ATPase inhibition by ATP6AP1KO or ConcA causes intracellular acidification, which increases mitochondrial membrane potential and improves cell fitness under ACT treatment (Previous manuscript, Page21 line 500-501 “These data suggest that intracellular acidification through inhibition of v-ATPase function improves cell fitness under mitochondrial dysfunction”). However, in the revised manuscript, making the intracellular environment more alkaline (or near neutral) to improve cell fitness is important. For example, Fig. 3H in the revised manuscript suggests that ACT treatment shifts the intracellular environment to acidic conditions, while ATP6AP1KO-ACT treatment conditions return it to the alkaline (neutral) side. These results are completely different from the claim of the previous manuscript. Particularly uncomfortable is that Fig. 5H in the previous manuscript shows a significant increase in intracellular pH in ATP6AP1KO compared to WT, however in the revised manuscript, Fig. 3H, there is no apparent difference between WT and ATP6AP1KO in intracellular pH. These results should be the same conditions, so an

explanation is required as to why this difference occurred. In addition, the claims in the revised manuscript do not explain how the loss of ATP6AP1 ameliorates the reduction in cell fitness caused under ACT treatment.

Reviewer #4

(Remarks to the Author)

Reviewer #5

(Remarks to the Author)

The revised manuscript has improved, though its complexity remains high, but the authors have largely addressed this Reviewer's comments. However, this Reviewer would agree that particularly comments by Reviewers #2 and #3 need to be addressed.

Reviewer #6

(Remarks to the Author)

Version 2:

Reviewer comments:

Reviewer #1

(Remarks to the Author)

In the revised manuscript, the authors adequately addressed my concerns.

Reviewer #2

(Remarks to the Author)

This revised version of the manuscript is written more clearly and has satisfied most of my previous comments. The major message of the paper is that partial inhibition of the lysosomal V-ATPase can compensate for mitochondrial defects arising from mistranslation, and the authors have proven this point. However, the data indicate that this compensation depends on a relatively small increase in lysosomal acidification and there are still some places where the wording does not convey this. I would recommend only the following text changes:

- 1). Lines 353-354, 406 and 559: Change "V-ATPase deficient" and "V-ATPase disruption" to partial loss of V-ATPase function or partially disrupted V-ATPase function.
- 2). The use of "low V-ATPase inhibition" and "low lysosomal defect" on lines 462, 464, 504-505, 530, 538-539, 543, 546, 549 are potentially confusing. I would recommend "partial V-ATPase inhibition" or "partial lysosomal defect" because this appears to be what is happening with very low concentrations of concanamycin/bafilomycin or ATP6AP1 KO.
- 3). Lines 521-523 which are in a revised section of the discussion indicate that "enlarged, less acidic lysosomes promote a shift to catabolism...during energy stress". However, recent data indicates that energy stress can also induce a shift to more perinuclear and more acidic lysosomes (Ebner et al. (2023) Cell 186:5328). This reference is also relevant here.

Reviewer #3

(Remarks to the Author)

In this revision, the authors have added quantitative experiments using glucose tracing and BCECF-AM, which significantly enhance the reliability of the data. As a result, some interpretations differ from those in the original manuscript; however, the central concept of "extra-organelle buffering" is now more robust and convincing, supported by substantial data. The manuscript's structure has also been simplified and clarified, improving readability.

I believe no additional experiments are necessary. Nevertheless, to aid readers' understanding, some sections require explanation or clarification as described below.

Minor points to be clarified

1. Regarding the activation of the PPP pathway, I understand that the authors did not examine the NADP/NADPH ratio due to concerns about quantitative reliability. Instead, in the new experiments presented in Figure 4D, glucose tracing using C¹³ was performed. This experiment is elegant, but the explanation is somewhat limited. While the manuscript clearly describes glucose uptake and lactate synthesis via the PPP pathway, it is unclear how TCA intermediates and amino acids (AAs) were quantified. The y-axis in the figure is not fully explained. Does it represent relative abundance, fraction labeled, or

another measure? This information should be explicitly stated, either in the figure legend or in the main text. The “Methods” section details the experimental procedures but does not describe the calculation methods.

2. Additionally, rather than combining all these results into a single Figure 4D, it may be preferable to present them as separate panels (Figures 4D, E, F, G, and H) with detailed explanations in the main text.

3. Regarding the glucose tracing figure, it appears that the red and blue circles both represent C¹³ labeling. If so, is there a specific reason for using different colors? Initially, it may be interpreted as indicating distinct molecules. Clarifying this in the legend—for example, with a note such as “filled circle indicates C¹³ position in glucose”—would be helpful.

4. In Figure 4D, for ATP6AP1 KO, “KO” should be written as a superscript.

5. Are GADP and G3P distinct substances? If so, it would be helpful to clarify their identities.

Reviewer #4

(Remarks to the Author)

We thank the reviewers for their positive feedback and helpful suggestions in improving our study and manuscript and address their remaining concerns and comments below. All alterations in the manuscript are highlighted in yellow in the revised version. Please note that not all figures in the responses to the reviewers are included into the revised manuscript.

REVIEWER COMMENTS

Reviewer #1 (Remarks to the Author):

Comments to the authors

In this manuscript, through CRISPR/Cas9 screen, the authors identify that the inhibition of vacuolar v-ATPase can restore the cell fitness under mitochondrial translation inhibitors-treated conditions. The study contains several interesting findings. However, this reviewer feels the current version of the manuscript is not easy to follow everything for several reasons. Also, some unclear points were observed. Please see specific comments below.

We thank the reviewer acknowledging the interest and novelty of our study and hereby address their concerns below.

Major points

1. Each figure panel is too small to read. For example, Fig. S5 contains so many information, and thus it is very hard to capture the main argument of the figure. I would suggest the authors to make each figure more understandable.

We aimed to increase readability and figure panels. To convey our findings more clearly, we have shortened the main figures and extended supplementary data.

2. The authors should focus on simple but sufficient description as much as possible to avoid misunderstanding of readers. For example, "CAP had no added effect on growth in ACT conditions but altered overall protein expression and lead to partial rescue of mitochondrial fragmentation with near to complete loss of translation at 5 µg/mL (Figure 1D; S1A, B)" on page 6. Several observations are described in one sentence. I think "mitochondrial fragmentation" is not properly addressed in indicated figures, although I guess the authors mention the OPA1 band pattern as an indicator of mitochondrial morphology. It is better to explain what kinds of protein expression are altered. Also on page 6, following sentence needs more simplified understandable description: "All drug concentrations were adopted to comparable growth inhibition to avoid subsequent proliferation-induced hit biases - prominent with loss of mitochondrial genes -, whilst ensuring sufficient resolution of the screen at low dose (Figure 1D)". Please notice that these are just part of examples.

We have aimed to increase readability by simplifying sentences and using clear terminology (e.g., OPA1 processing as a proxy of mitochondrial fragmentation).

3. The purpose and the interpretation of the series of experiments in Fig. S3K-O are not clear.

We agree with the reviewers' comment regarding the description of the respective purpose of the listed experiments. This experimental series was initially conducted to dissect particularities of our model. For clarity, we have removed unnecessary descriptive data of this experimental series.

4. From several analyses in Fig. 4 and Fig. S5, the authors concluded "v-ATPase-dependent deficiency remodels cellular metabolism beneficial under mitochondrial dysfunction". But it is not clear what kinds of beneficial metabolic changes are actually induced, and how those are linked with the observed restored mitochondrial and cellular fitness. In Abstract, the authors mentioned that "v-ATPase regulates the cellular redox environment", but is this proved in this figure? If not, where is the evidence?

The conclusion of v-ATPase deficiency being beneficial under mitochondrial dysfunction in our model is related to rescued growth and therefore a direct measure of cell fitness. We agree that referring to general metabolic benefits in this respect are not sufficiently explained. We therefore clarified terms and provided additional pathway analyses to characterize how their respective metabolism is rerouted (new Figure 4). We also altered the term "redox environment" to "pH homeostasis".

5. OPA1 is known to be cleaved by OMA1 in response to mitochondrial depolarization. In Fig. S6A, this does not seem to be induced even in WT cells.

This is correct. Indeed, short-term treatment (<24h) with the commonly applied concentrations of ionophores FCCP/CCCP depolarize mitochondria and induce OPA1 processing (see our additional data Figure R1 with increasing concentrations of FCCP and Anand et al., 2014 (doi: 10.1083/jcb.201308006)). Extended treatments with these concentrations will result in cell death.

In order to mimic persistent mitochondrial dysfunction as a concept of "disease progression/severity", we titrated most inhibitors in our models. In the previous Figure S6A OPA1 levels are from the respective 3-days treatments in cells and retain significant growth under low FCCP concentrations. We hypothesized that the higher membrane potential in the ATP6AP1^{KO} will render them less sensitive to FCCP in comparison to WT cells. The data in previous Figure S6A is in supports of this. For clarity, we have specified and indicated the respective treatment times in the revised manuscript and show population doublings in the current figure (Figure S4A).

New Figure R1: Effect of short-term exposure to increasing concentrations with FCCP on OPA1 processing in WT HEK293 cells. 24 hours treatment with 20-60 μ M FCCP. The dotted line represents where the image was cut to show FCCP only.

6. In Fig. S6B, the authors mentioned that v-ATPase inhibition increases OPA1 expression. But in Fig. 3F, the authors argued that ATP6AP1 deletion (kind of v-ATPase inhibition) prevents the OPA1 processing. Which is the authors' conclusion?

In our ATP6AP1 deletion model we do observe altered OPA1 processing and as a result increased abundance of long OPA1 isoforms. We have altered the wording in the manuscript to clarify this. We performed treatments of concanamycin A on WT HEK293 cells showing that v-ATPase inhibition does not increase overall OPA1 protein levels or cause OPA1 processing in the concentrations assayed (**Figure R2**).

WT concA treatment

New Figure R2: Effect of v-ATPase inhibition/deletion on OPA1 expression and processing. Low-dose (50-400pM) for three days or 30 minutes high-dose treatment (10nM) with the v-ATPase inhibitor concanamycin A in WT HEK293 cells.

7. In Fig. 5E and 5F, the authors focused on extracellular pH. But the underlying logic that leads to this experiment is not explained clearly.

Whilst our ATP6AP1 cells are more sensitive to acidic extracellular media, we did not assess how this affects intracellular pH in our models. In our revised experiments, we show that mitochondrial dysfunction by ACT lowers the cytoplasmic pH and the ATP6AP1 cells are able to buffer this more efficiently (**new Figure 3**).

8. In Fig. S6G-I, the authors showed that ATP6AP1 KO cells are also resistant to staurosporine or FCCP-treated conditions. Does this mean the beneficial effects of v-ATPase inhibition are not specific to mitochondrial dysfunction that is caused by mitochondrial translation inhibitors?

Because the ATP6AP1^{KO} cells display an increased mitochondrial membrane potential compared to WT cells, we hypothesized them to be more resistant to increasing concentration of the mitochondrial membrane potential-dissipating ionophore FCCP. Staurosporine-induced apoptosis has been shown to trigger loss of mitochondrial membrane potential in comparable concentrations (1-10nM; Scarlett et al., 2000; doi: 10.1016/s0014-5793(00)01681-1), prompting a similar conclusion as with FCCP in that their resistance is related to their increased mitochondrial membrane potential. The insertion of aberrantly translated mitochondrial protein induces loss of mitochondrial membrane potential and cells with higher membrane potential therefore are expected suppressors of our condition. Whilst we cannot exclude further beneficial effects of v-ATPase inhibition, these cells show comparable sensitivities for other inhibitors (previously Fig. S3K-O, see point 3.).

9. Higher mitochondrial membrane potential in ATP6AP1 KO cells is very clearly presented (Fig. 5A). This is a key finding of this study. However, the underlying molecular mechanisms are not described well in the current study.

We have now clearly emphasized that the consequence of insertion of aberrantly translated mitochondrial proteins into the inner mitochondrial membrane results in mitochondrial membrane dissipation. Cells with increased MMP, therefore are potential suppressors in our screen. The ATP6AP1 cells clearly show a higher MMP and are less susceptible to ionophores such as FCCP. We have extended our experiments (new data in Figure 3 and 4) and refined our working model.

Minor points

1. Same immunoblot panels seem to be used in Fig. 1B and Fig. 2A. At least, the authors should mention this in the figure legend, if that is the case.

This is correct. We decided on this "duplication" to introduce the reader to the screening outline for mis- and non-translation (Fig.1B) before adding the data on mitochondrial ribosomal integrity. For clarity, we have removed this duplication in the revised version.

2. Fig. S3J is not mentioned in the text.

We apologize for this mistake. We have removed this data to improve clarity.

3. Several panels in Fig. S6 are not properly cited in the text: panels E and F are not mentioned, panels O and P do not exist.

We removed data of staurosporine-induced cell death and cleaved PARP as these experiments showed comparable susceptibilities in WT and KO and are not essential to our conclusions. In addition, we also split the supplementary data pertaining to our U2OS cell model.

Reviewer #2 (Remarks to the Author):

In this manuscript, the authors seek to understand the cellular consequences of mistranslation of mitochondrial proteins through a genome-wide CRISPR screen in the presence of actinonin, an inducer of mistranslation, with and without the mitochondrial translation inhibitor chloramphenicol. In addition to several expected hits, genes against several V-ATPase subunits emerged as potential suppressors of actinonin effects on mitochondria. The authors choose to focus on effects of ATP6AP1, also known as Ac45, a V-ATPase assembly factor that was shown to be part of the lysosomal V-ATPase in recent structures. Extensive characterization of mitochondrial structure, mitochondrial proteins such as Opa1, and the overall metabolic state of cells in wild-type and ATP6AP1 knockout cells are provided. Unfortunately, many of these experiments are either not tied back to mechanism or provide few insights into the mechanistic consequences of mistranslation and how reduced V-ATPase activity might compensate for the effects of actinonin. In their final analysis, they conclude that the ATP6AP1 knockout or low concentrations of the inhibitors concanamycin and bafilomycin may alter overall pH homeostasis, lowering cytosolic pH and therefore increasing mitochondrial membrane potential. This is a reasonable hypothesis for a potential effect from V-ATPase activity, but it is not conclusively tested here. As a result, the authors have a potentially interesting result from their screen, but it provides no real mechanistic insight.

We thank the reviewer for acknowledging their interest in our results and have addressed their concerns below. We would like to emphasize that treating WT cells with low concentrations over a period of 3-days with low amounts of the well characterized inhibitors concanamycin A or bafilomycin A alone is sufficient to increase the mitochondrial membrane potential. We extended our experiments to primary cell models of patients demonstrating that low v-ATPase inhibition alone improves cell growth and mitochondrial membrane potential. We have performed pH-calibrated probe and biosensor assays and are convinced that the supporting experiments will substantiate the conclusions in our revised manuscript.

The following points should be noted if the paper is revised.

1. The authors report that their ATP6AP1 knockout is hypersensitive to V-ATPase

inhibitors relative to wild-type cells (lines 290-292). This result is disturbing, as it indicates that there must still be residual V-ATPase activity somewhere in the cell, since these inhibitors are highly specific. This key point is not addressed. It is possible that the knockout only affects a subpopulation of V-ATPases that require ATP6AP1 or else the ATP6AP1 knockout is not complete. Either possibility complicates interpretation of most of the subsequent results.

As the reviewer notes, highly specific v-ATPase inhibitors were used in the present study to independently phenocopy the results obtained in the ATP6AP1 knockout cells. These inhibitors target all v-ATPases within the cell as both drugs are specific to the V₀ subunit c (Huss et al., 2002; doi: 10.1074/jbc.M207345200). We have used sublethal concentrations of the v-ATPase inhibitors in WT cells and show increased mitochondrial membrane potential (Figure 5C). Our data show that both genetic manipulation (knockout of the subunit ATP6AP1 subunit) or incomplete pharmacological inhibition with well-characterized v-ATPase inhibitors rescues cell fitness in different cell types. We mention in the text that *“ATP6AP1 knockout (KO) cells were highly sensitive to the commonly used v-ATPase inhibitors bafilomycin A1 (bafA) and concanamycin A (conca) at low concentrations not affecting cell growth in WT cells (Figure 3D), suggesting impaired but residual v-ATPase function is essential for cellular viability”*. v-ATPases are essential enzymes with diverse physiological roles. In the fractionation experiment we show that ATP6AP1 is predominantly associated with lysosomes and as such an essential component of a lysosomal signaling hub in HEK293 that functions to balance anabolic and catabolic processes which cannot be fully inhibited (Collins et Forgeac, 2020. doi: 10.1016/j.bbamem.2020.183341).

We have extended our data to show the physiological relevance of low-dose inhibition (titrated to 40pM conca) of the v-ATPase in patient-derived fibroblast cell lines. Commonly used concentrations of concanamycin (10nM conca) show that v-ATPase activity is essential for cell growth/viability. We have now added this data to the revised manuscript (new data for Figure R3). In contrast, the residual activity of ATP6AP1^{KO} phenotypically mimics low-dose v-ATPase inhibition with concanamycin A in HEK293 and U2OS cell lines.

Several conclusions can be drawn based on our data:

- 1) Loss of ATP6AP1 shows increased sensitivity to v-ATPase inhibitors confirming that our knockout is indeed affecting v-ATPase function.
- 2) We have validated our ATP6AP1 antibody in two independent cell lines, namely in our ATP6AP1 knockout clones in HEK293 and U2OS cells (Figure R4).
- 3) The ATP6AP1 antibody also recognizes exogenous expression of our ATP6AP1-FLAG construct, confirming the specificity of the antibody used to confirm the knockout.
- 4) The localization of a transient expression of an ATP6AP1-FLAG construct does suggest predominant vesicular distribution (new data in Figure R5).

- 5) We have performed ATP6AP1-FLAG immunoprecipitation experiments to identify potential interactors and to allow for conclusions on localization (new data in Figure R6). These results show that ATP6AP1 interacts as expected predominantly with V_0 subunits (e.g., ATP6AP2, ATP6V0A1-3, D1-2) and lysosomal proteins where most v-ATPase is located.
- 6) The v-ATPase inhibitor concanamycin A phenocopying the ATP6AP1^{KO} is specific to all v-ATPases.
- 7) ATP6AP2 levels are almost undetectable in our ATP6AP1^{KO} cell line, in agreement with proteomic data of v-ATPase subunits (new data in Figure R7). These proteins are interacting and in close proximity within reported v-ATPase structures (Abbas et al., 2020; doi:10.1126/science.aaz2924) and undetectable levels of ATP6AP2 suggest complete knockout of ATP6AP1.

In conclusion, our experiments confirm the knockout of ATP6AP1, but that its loss does not result in complete ablation of v-ATPase activity. Whilst we agree with the reviewer that incomplete knockout could serve as an argument, using two highly specific drugs resulting in an increase in cell growth at low concentrations (in contrast to high – Fig. R3) under mitochondrial mistranslation, are in our opinion convincing. Importantly, we now show that low-dose v-ATPase inhibition alone in primary fibroblasts of patients with mitochondrial translational phenotypes are sufficient to increase cell fitness/ growth and mitochondrial membrane potential (new Figure 5; new Figure R16).

New Figure R3: Cell growth rates in HEK293 WT cells over a period of 3 days with indicated concentrations of concanamycin A. Dots represent independent experiments.

Figure R4: Immunoblotting against ATP6AP1 for knockout confirmation in HEK293 and U2OS.

New Figure R5: Immunofluorescent staining against anti-FLAG and anti-ACTIN in HEK293 WT cells transiently transfected with an ATP6AP1-FLAG construct.

New Figure R6: Immunoprecipitation coupled with protein identification and quantification by mass spectrometry showing the detected subunits of the v-ATPase using ATP6AP1-FLAG as bait (new PRIDE project data accession: PXD057148; token: AVzqoTwKIPMS).

New Figure R7: Immunoblotting against ATP6AP2 in WT and ATP6AP1^{KO} cells. In agreement with proteomic results, ATP6AP2 protein levels are almost undetectable in the ATP6AP1^{KO} cells. Asterisk indicates unspecific band form LAMP1 incubation.

2. In Figure 4G and H, the authors show that the ATP6AP1 knockout still accumulates LysoTracker in acidic compartments, and may in fact accumulate more. This is inconsistent with the expected loss of vacuolar acidification in this mutant. The presence of other core V-ATPase subunits in the original CRISPR screen would suggest that loss of V-ATPase activity and acidification is what suppresses effects of actinonin.

We fully agree with this and have extended our analysis for acidic compartments (also in accordance with reviewer #3). We have now performed pH-calibrated measurements using a ratiometric protein-based pH sensor of lysosomal luminal-oriented TFP1 and extralysosomal mCherry demonstrating alkalinisation in our v-ATPase mutant (new Figure 3, data in Figure R8). Whilst lysoTracker stains acidic compartments, we used it fixed to quantify our experiments regarding particle size and number (see new data in Figure R12).

New Figure 3F/R8: Determination of lysosomal pH using a ratiometric FIRE-pHLy (Chin et al., 2021; doi: 10.1021/acssensors.0c02318) in WT and ATP6AP1^{KO} cells treated with ACT.

3. The possibility that perturbed pH homeostasis might be responsible for the apparent suppression of actinonin effects is reasonable but is not tested thoroughly. Both lysosomal pH and cytosolic pH could be determined quantitatively with ratiometric pH sensors. However, even labels like pHrodo-green that are not ratiometric can be calibrated to pH. Simply reporting mean fluorescence intensity as in Figure 5H is not adequate.

We agree with the reviewer's notion regarding a more sophisticated methodological approach. We have used different experimental protocols and dyes with respective pH calibration to report actual values in our experimental settings.

New Figure 3H/R9: Determination of cytoplasmic pH using pH-calibrated pHrodo in WT and ATP6AP1^{KO} cells treated with ACT.

4. There is no question that the authors did a lot of work for this paper, but the Figure legends could be improved to make it easier to determine exactly what is being shown.

We thank the reviewer for acknowledging our work and their valuable feedback. We have aimed to improve readability and clarity in the revised manuscript and have improved our study significantly based on their experimental suggestions.

Reviewer #3 (Remarks to the Author):

Using a genome-wide CRISPR screen, the author identified H⁺-type vacuolar (v-)ATPase subunits as regulators of mitochondrial dysfunction by disrupting the inner mitochondrial membrane. The author elucidated the mechanism by which v-ATPase rescues mitochondrial dysfunction and introduced the concept that lysosome function can regulate mitochondrial membrane potential. This work comprises a substantial amount of data and is interesting in terms of inter-organelle communication. However, the following issues need to be addressed before publication.

We thank the reviewer for the concise summary of our work and the reference to inter-organelle communication. We have addressed their comments below. We also would like to thank the reviewer for the helpful experimental suggestions to improve our study and further confirm our findings.

Major Comments

1. PPP Pathway Activation Evidence: In Figures 4A-C, the author claims that the pentose phosphate pathway (PPP) is activated in actinonin (ACT)-treated v-ATPase-deficient cells due to a decrease in total amino acid concentration. If the PPP is activated, the NADP/NADPH ratio should decrease, and the GSH/GSSH ratio should increase. However, in Figure 4B, the GSH/GSSH ratio decreases in ACT-treated v-ATPase-deficient cells compared to non-treated v-ATPase-deficient cells. To substantiate the claim of PPP activation, the NADP/NADPH ratio should be measured in addition to GSH/GSSH. Additionally, performing a label-free proteome analysis to examine changes in protein levels involved in glycolysis (e.g., ALDOA, PKM2, LDHA) and the PPP (e.g., G6PD, PGD) under each cell condition would support this hypothesis.

We thank the reviewer for their suggestion. We point out that our ratios from untargeted metabolomics are a crude approximation and reflect a steady-state and no flux. The evidence for PPP activation was linked to the increase in PPP-relevant metabolites (see R-5-P/PentoseP, HMDB000061; **new data in Figure R10**). However we did not see alterations in the expression of key PPP enzymes in our proteomic data. We altered the wording of this passage in the revised version of the manuscript.

New Figure R10: PPP activation, label-free proteome analysis of PPP enzymes (detected 16/22 in KEGG PPP, dots are the mean of quadruplicates of each detected PPP protein)

and PPP-related metabolites in treated and untreated WT and ATP6AP1^{KO} cells.
Scheme adapted from TeSlaa et al., 2022; doi: 10.1038/s42255-023-00863-2).

2. Sample Consistency in Fig. 4C: Clarification is needed regarding the sample differences between "WT ACT" in the upper panel and "ACT" in the lower panel of Fig. 4C. If both data sets represent normal HEK293 cells treated with ACT, the observed differences require an explanation or rectification.

We thank the reviewer for spotting this typo, which we have now corrected.

3. LAMP2 Expression in Fig. 4F: The claim of increased LAMP2 expression in ATP6AP1KO cells in Fig. 4F appears unclear. This should be substantiated by also presenting LAMP2 levels from the label-free analysis in Figure 4D for better clarity and support.

Label-free proteomics did not identify LAMP2. We do observe a retention in the gel of both LAMP1 and 2 in the KO compared to WT, likely due to their altered glycosylation. We have changed the text to emphasize this and not their abundance (new data in Figure R11).

Figure R11: Western blot analysis of LAMP1/2 in WT and ATP6AP1KO cells treated with ACT. Independent experiments.

4. LysoTracker Signal Interpretation in Fig. 4G and 4H: Figures 4G and H show an increased lysoTracker signal, indicating more acidic vesicles. Since lysoTracker signal intensity is pH-dependent, this could result from either an increase in the number/volume of acidic vesicles or a decrease in lysosomal pH, or both. Analyzing the number and volume of lysosomes per cell in Fig. 4G and discussing the metabolic implications of these changes would provide a more comprehensive understanding.

We thank the reviewer for their suggestion and have now included quantitative values of our imaging (new data Figure R12 and additional pH-calibrated experiments in Figure 3/R8) and extended the discussion of these alterations as suggested by reviewer 2.

F

New Figure 3E, 8D/R12: Quantitative analysis of average acidic vesicle size.

5. LAMP1 and ATP6AP1 Co-Migration in Figure S4G: The co-migration of LAMP1 and ATP6AP1 in subcellular fractionation (Figure S4G) suggests potential interaction. If an interaction is assumed, the authors should discuss its significance and potential functional implications.

The whole-cell digitonin-based permeabilization assay is crude and would not allow for direct interaction. We have performed additional immunoprecipitation experiments using ATP6AP1-FLAG to identify potential interactors with several lysosomal proteins enriched and expectedly v-ATPase subunits, which are now included in the manuscript for reference and new data in Figure R6.

6. Intracellular pH and Mitochondrial Membrane Potential: The authors suggest that ACT treatment decreases membrane potential, and v-ATPase perturbations increase cellular pH, thus recovering mitochondrial membrane potential. Testing intracellular pH changes in ACT-treated cells using TMRM and examining intracellular pH under the conditions shown in Fig. 5B (where mitochondrial membrane potential increases in ATP6AP1KO cells treated with ACT) would validate this hypothesis. An explanation for the increased mitochondrial membrane potential under these conditions is also required.

We thank the reviewer for their valid hypothesis and have performed additional experiments including the pH-calibrated measurements with the conditions shown in Fig5.B (New figure R8/R9). Our conclusion from these experiments is that the suppression of the actinonin effect is related to an improved buffering capacity in the knockout cell line. We extend on the explanations for increased mitochondrial membrane potential in the discussion.

7. pH Buffering and Cell Fitness in ACT-Treated Cells: To show the importance of pH buffering for cell fitness in ACT-treated cells, the authors should modify pH by adding HCl or NaOH to the medium and assess cell fitness. Additionally, examining other parameters such as OPA1 processing, changes in protein levels, mitochondrial fragmentation, and rerouting of glycolysis in v-ATPase-deficient cells would provide further insights.

This is an important contribution. Acidification of the extracellular medium collapses the mitochondrial membrane potential in the ATP6AP1^{KO} cells and overall treatment with either concentration of HCl or NaOH do decrease cell growth in WT and KO. From our new data, we conclude that the alkalinization is beneficial under conditions of mitochondrial mistranslation and that the intracellular pH homeostasis is buffered in the KO cells. Whilst our knockout model is more sensitive to an acidic extracellular medium subsequently decreasing MMP (Figure S5D-F/R13), merely decreasing extracellular media under ACT treatment did not rescue cell fitness in either condition (new data in Figure R14). This suggests that changing extracellular pH alone is not sufficient to alter intracellular pH to improve cell fitness in WT cells. Furthermore, the concentrations that we have used for this assay (20mM HCL or NAOH) affecting MMP, are also decreasing cell fitness significantly, which could explain the additive effect of ACT and extracellular pH alteration on cell fitness.

Figure R13: Impact of alteration of extracellular pH on cell fitness and MMP.

New figure R14: Impact of alteration of extracellular pH on cell fitness under ACT treatment in WT cells.

Minor Comments

1. Western Blotting in Fig. 1B and Fig. 2A: If the same western blotting results are used in Fig. 1B and Fig. 2A, this should be explicitly stated.

This is correct and we have now clearly indicated this. We decided on this "duplication" to introduce the reader to the screening outline for mis- and non-translation and OPA1 processing first (Fig.1B). We have removed this duplication in the revised manuscript.

2. Character Size in Fig. 2F and G: The character size in Fig. 2F and G is too small to read, and the information is not informative enough. These figures should be omitted or revised for clarity.

We have revised these panels for clarity.

3. Label-Free Quantitative Proteomics in Fig. 3E: Clarify if label-free quantitative proteomics detects ATP6AP1/2 even in ATP6AP1KO conditions as shown in Fig. 2C. Also, clarify the Y-axis value and what was normalized and set to 1.0.

Our proteomics data shows that ATP6AP1 was clearly identified from all WT samples with high intensity and good quality MS/MS data. ATP6AP1 was also identified with low intensity from some of the KO samples, but only when using 'match between run' (MBR) function in MaxQuant database search. MBR is designed to increase the number of identifications from very low abundant proteins from the samples where only low abundant peptide ion signal can be detected from the sample and identification is based on comparing to MS/MS data from the other samples analysed in the same MaxQuant search. Since ATP6AP1 was identified from all WT samples with high intensity it is possible that the MBR-based identifications in KO samples are carryover signal from WT samples. We have now clarified values and normalization in the revised manuscript.

4. Figure Legend Discrepancy: On page 13, line 308, the text describes "72 h ACT-treated WT cells" while Figure S3B indicates "15 days treatment." Verify and correct this discrepancy. Additionally, ensure that references to Figure 3E and Figure S3B are accurate.

Figure S3B is correct as it represents actual label-free proteomic data from a 15-days low-dose ACT treatment (the conditions of the actual screen). Subsequent specific experiments in WT and KO cell lines were performed over a time course of 3 days. We present this data for comparability of the screen conditions and our other time points. For clarity we have removed this data.

5. Missing Labels in Fig. 4C: The label "ACT" is missing from the right bottom, right side of Fig. 4C. This should be added.

Thank you. This has been corrected.

6. Unexplained Figures S3J and S4B: Figures S3J and S4B are not explained in the manuscript. These should either be explained or removed.

We have explained, clearly labelled or removed the respective figures.

7. Typographical Error in Fig. 5G: Correct the typo "Phrodo" to "pHrodo" in Fig. 5G.

This has been corrected.

8. Figure Reference Correction: On page 22, line 505, correct the figure reference from (Figure S6J-P) to (Figure S6J-N).

This has been corrected.

9. Terminology Correction: On page 23, line 526, change "inter-organelle" to "extra-organelle."

This has now been redacted and consistently named "extra-organellar".

Reviewer #4 (Remarks to the Author):

We appreciate serving as example for peer-review training.

Reviewer #5 (Remarks to the Author):

In this manuscript, entitled 'v-ATPase-mediated extra-organellar buffering resolves mitochondrial dysfunctions', the authors propose a new signaling mechanism where mitochondrial translation-induced stress is alleviated by the inhibition of the vacuolar (v-)ATPase, a complex responsible for the acidification of intracellular compartments. Methodologically, the authors first apply a genome-wide CRISPR screen in the presence

of mitochondrial ribosome inhibitors to identify pathways/modulators of cellular fitness upon which the v-ATPase complexes emerged as an interesting candidate. Following a wide array of cell biological and biochemical characterizations, including in knockout lines, the authors describe v-ATPase as a regulator of mitochondrial cristae morphology and mitochondrial membrane potential and suggest v-ATPase inhibition to be beneficial in specific conditions of mitochondrial dysfunction, with broader implications, including to cancer and aging.

We thank the reviewer for their excellent summary and recognition of the potential broader implications.

Generally, identifying mechanisms of mitochondrial-nuclear crosstalk and adaptation to cellular stress responses is of broad interest given the breadth of conditions, where mitochondrial dysfunction has been described, including in primary mitochondrial disorders. Overall, the experiments appear to have been thoroughly done and this reviewer did not identify any obvious shortcomings concerning their execution. However, the following comments should be strongly considered to improve the manuscript.

We thank the reviewer for highlighting the appropriate experimental conception and acknowledging our methodological approach.

Major comments

1. Clarity - The manuscript is very densely written and has an extremely high degree of complexity, which will make it challenging for a general audience to appreciate. This reviewer would suggest revising the text for clarity to better follow the author's reasoning and decision-making of their experimental work-up.

We have improved clarity to make our observations more accessible to a general readership. We have extended the supplementary data and excluded non-essential data in our revised version.

2. For most of the manuscript, the authors conduct experiments in 293 cells that are certainly adequate for the screen and subsequent biochemical characterizations. Cancer-derived U2OS cells are then used as a form of validation, although the scope of related experiments is relatively quite limited. In addition, the cell lines appear to be mostly conducted in the presence of high glucose, raising the overall question of how generalizable the described findings are considering the framing surrounding primary mitochondrial disorders, neurodegenerative diseases, and cancer. Specifically, can the authors confirm the buffering effect of v-ATPase-mediated in a primary cell model, including with more physiologic media conditions (e.g., human plasma-like media)?

The reviewer raises an important general point regarding cell culture media and the cell lines chosen as well as generalization or applicability to physiological models. In respect to this point, we aimed to confirm our screen findings in another cell line (U2OS). We

have now also used primary patient-derived cell models to extend the applicability of our findings in physiologically relevant models for mitochondrial dysfunction (see point 3).

Based on the reviewer's suggestion, we have performed additional experiments in low glucose and galactose conditions in our models showing that lower glucose availability exacerbates mitochondrial dysfunction in WT cells, but not in our dysfunctional v-ATPase model (ATP6AP1KO). We have now added this data to the revised manuscript (new Figure 4B/R15).

New Figure R15: Altered glucose availability exacerbates mitochondrial dysfunction in WT, but not in v-ATPase deficient cells.

3. Along these lines, mitochondrial dysfunction in vivo may arise in response to a variety of stressors, genetic mutations in nuclear as well as mtDNA affecting protein-coding genes as well as tRNAs, hypoxia, nutrient deficiency, etc. As of the current state, it remains unclear to what extent the described pathway is of broader relevance, which is something the authors also partially elaborate on in their discussion. The author is not necessarily asking for additional experiments (which of course would also be appreciated), but the limitations in terms of the interpretation of their results should be stated very clearly.

We agree with the reviewer's comment on the heterogenous origin of mitochondrial dysfunction. The particularity of mitochondrial mistranslation is also a consequence in mtDNA deletion disorders, where potentially aberrant mt-mRNA species on mitoribosomes induce mistranslation. Indeed, this is exemplified in our screen using actinonin resulting in a comparable stress response in mice with mtDNA deletions (Forsstrom et al., 2019). We have included limitations on generalization of this mechanism to particular mitochondrial pathomechanisms.

In the revised version of our manuscript, we now present that v-ATPase buffering improves cell growth in the primary cell model of patient-derived fibroblasts cell lines

with particular mitochondrial translation defects. We demonstrate that low v-ATPase inhibition alone is sufficient to increase cell fitness/growth in particular models of mitochondrial mistranslation, confirming our findings in physiologically relevant cells and patient-relevant mutations. We have now added this data to the revised manuscript (new Figure R16).

New Figure R16: Concomitant v-ATPase deficiency rescues cells growth in pharmacological and patient-derived primary fibroblasts models of mitochondrial mistranslation.

Reviewer #6 (Remarks to the Author):

We appreciate serving as example for peer-review training.

REVIEWER COMMENTS

Authors: We would like to extend our sincere appreciation to all reviewers for their constructive comments contributing to enhancing the scientific rigor and quality of this study, their willingness to re-evaluate our revised manuscript, their overall acknowledgment that our study was improved aided by their suggestions and the consensus that mild v-ATPase inhibition confers resistance to mitochondrial dysfunction. In our revised manuscript, we have added experimental data using state-of-the-art ratiometric pH-calibrated measurements and metabolic tracing as orthogonal approaches. We also clarified remaining concerns and clearly indicated any alterations made line by line in this point-to-point response, which are highlighted in yellow in the revised version of our manuscript.

Reviewer #1 (Remarks to the Author):

Comments to the authors

In the latest version of the manuscript, the authors have made substantial edits on text and figures. The content is now much improved to follow. The study is extremely dense and comprehensive. However, this reviewer feels that it lacks necessary experiments to support their main argument. I also feel that it is sometimes difficult to accept the authors' interpretation of data. I fully agree that mild inhibition of lysosomal V-ATPase function has beneficial effects on cell fitness when cells are encountered with mitochondrial stress. However, some keywords such as "organelle communication" and "retrograde signaling" may be somewhat overstatements, since several key phenotypes in ATP6AP1 KO cells identified here (for example, higher mitochondrial membrane potential indicated in Fig. 2G) are steady-state phenotypes, and the mitochondrial stress-dependent active crosstalk between the two compartments was not identified in this study. In summary, the study contains a lot of information and potentially important findings, but the authors are encouraged to reshape the overall content to support their main argument with solid data. Please see specific comments below.

Authors: We appreciate the reviewer feedback and agree on inherent restraints in the use of knockout models that the observed phenotype might not stem from a direct crosstalk and is adaptative in nature, thereby representing "steady-state". We therefore would like to emphasize again that the use of a well-defined titrated inhibitor allows to unequivocally demonstrate a direct correlation and that this pharmacological inhibition of the v-ATPase phenocopies the impact on mitochondrial membrane potential and cell fitness when cells are encountered with mitochondrial stress, supported by experiments in Figures 5 D, E, F, G; S7 G; now S5 H-K). We agree that the active organellar crosstalk and retrograde signalling might be overinterpretation as we do not show direct spatial interaction and have therefore carefully revisited our statements and softened our conclusions (Abstract: line 29). The main argument of this study – as the reviewer acknowledges – is that low v-ATPase inhibition confers resistance to mitochondrial stress.

Major points

1. In Fig. 2F, the authors monitored OPA1 processing as an indicator of mitochondrial morphology (line 263-264). However, as shown in Fig. S3F, tubular mitochondria are significantly reduced in ATP6AP1 KO cells from the steady state, which means the OPA1 processing is not a good indicator of mitochondrial fragmentation in this context. Therefore, the authors' interpretation of these data (line 281-283) and the model illustration (Fig. 2I) are difficult to accept.

Authors: We agree with this comment and have specified this more clearly. While it is accepted that (ACT-induced) OPA1 processing is equivalent to mitochondrial fragmentation (Richter et al., 2013; DOI:10.1016/j.cub.2013.02.019), the observation of fragmented mitochondria does not imply or allow the reverse conclusion that OPA1 must be processed. In our particular case, the literature-adapted model illustration is valid, but we have clarified this implication of OPA1 as proxy for fragmentation clearly in the text (line 201-03) and in the model in Figure 2I (detailed in Figure S3D). We further confirm this by assessing the mitochondrial fragmentation state by immunofluorescence using the outer membrane protein TOM20 (now Figure 2F and S3F).

2. In Fig. 2G, the authors indicated that ATP6AP1 KO cells have a higher mitochondrial membrane potential already at the steady state. In Fig. 3H, the authors indicated that actinonin induces cytosolic acidification upon treatment in WT cells, but not in ATP6AP1 KO cells, which is an interesting observation. These two factors (mitochondrial membrane potential and intracellular pH) are important phenotypes observed in ATP6AP1 KO cells, and the authors try to relate them to the observed mitochondrial stress resistance in ATP6AP1 KO cells. However, several uncertain points remain in this argument. 1) Are these phenotypes related to each other? If so, what are the underlying mechanisms? – This point is related to my previous comment 9. 2) There is no direct evidence that proves these two factors indeed confer the stress resistance to ATP6AP1 KO cells. Regarding this point, as I indicated in my previous comment 7, I still don't understand the purpose of the experiments conducted in Fig. S5 D-G. To test that the cytosolic pH does indeed affect the mitochondrial stress resistance of ATP6AP1 KO cells, the experiments should be performed under actinonin treatment. For example, if the cytosolic pH is adjusted to acidic in ATP6AP1 KO cells, do these cells lose their resistance to actinonin?

Authors: We agree with the reviewer's point regarding the direct molecular connection of the phenomenon of mitochondrial membrane potential and cytosolic acidity. The observation that lowered cytosolic pH slows cell growth, and that pH reversal (alkaline intracellular vs. acidic extracellular pH) fuels cancer progression and induction of mitochondrial dysfunction decreases cytosolic pH have been established in previous studies (Johnson & Nehrke, 2010: doi: 10.1091/mbc.e09-10-08749). We agree with the reasoning that experimentally lowering the cytosolic pH should diminish the resistance of ATP6AP1KO cells to actinonin. The experiments the reviewer refers to (S5 D-G, now S5H-K) were undertaken for that exact reason. The results show that the KO in fact has a lower range of pH buffering capacity in both acidic and alkaline environments in regard to cell growth but sustains an alkaline cytosolic pH (Figure 3G). Measurement of the $\Delta\Psi_m$, however, shows that the resistance of ATP6AP1KO cells is increased because of their maintenance of $\Delta\Psi_m$, especially under alkaline conditions. Under acidic stress, ATP6AP1KO cells lose $\Delta\Psi_m$, suggesting that the v-ATPase is a key mechanistic player rewiring of pH homeostasis in the observed mitochondrial stress.

3. Based on the data set in Fig. S4 (and their response to my previous comment 8), the authors considered that ATP6AP1 KO cells are resistant to a wide range of mitochondrial stresses that induce mitochondrial membrane potential loss. Although the study was originally initiated by mitochondrial translation inhibitors, it is better to include these observations in the graphical abstract.

Authors: We thank the reviewer for their suggestion and have extended our observations in the graphical abstract to include resistance associated with stressors inducing mitochondrial membrane potential loss.

Reviewer #2 (Remarks to the Author):

In this revised manuscript, the authors have strengthened their evidence for a reciprocal relationship between mitochondrial membrane potential and morphology and lysosomal v-ATPase activity. The data are stronger, but some points in the text and figures need to be clarified. These changes should not require additional experiments but are important for the overall message of the manuscript.

Authors: We appreciate the reviewer acknowledging improvement of the manuscript and that the previous version was experimentally sound for the conclusions made. We have addressed their comments below and indicated the respective alterations clearly in the responses and the revised manuscript.

1. In Figure 3F, the authors compare calibrated measurements of lysosomal pH in normal and ATP6AP1 KO cells. These measurements show that lysosomal pH increases rather modestly, by 0.6 pH units in the untreated mutant cells and 0.3 pH units in the ACT-treated cells. This relatively small increase is consistent with the ATP6AP1 KO not completely abolishing v-ATPase activity in the lysosome, as acknowledged by the authors, but the Figure 3 legend states “its loss causes organelle alkalization” and the authors state on line 392 that “the ATP6AP1 KO cells had an alkaline pH”. A pH of 4.9 is not alkaline. Similarly on line 299 and 302 the authors cite “loss of v-ATPase”. A strength of the paper is the determination that low concentrations of v-ATPase inhibitors, which do not abolish activity, mimic the effects of the ATP6AP1 mutant, but the statements above confuse this comparison. It would be better to refer to reduced v-ATPase activity or partial loss of lysosomal acidification by ATP6AP1 KO throughout the paper.

Authors: We agree that “alkaline” is open to misinterpretation without referral to what it is related to. Based on the reviewer’s suggestion, we have altered any reference to “alkaline” to “more alkaline” or “partial loss of lysosomal acidification” in the relevant passages (lines Results: 237; 240; 250; 308-09; 363395; Discussion: 537). We use “loss of v-ATPase subunit ATP6AP1” in reference to our gene knockout.

2. Figure 3H shows that WT cells experience cytosolic acidification upon ACT treatment, while ATP6AP1 KO does not. This is an important result that deserves a little more discussion. It could indicate an issue with cellular buffering capacity as the authors suggest, but on the simplest level, a lower cytosolic pH (resulting to an increased ΔpH across the mitochondrial membrane) might actually increase, not decrease, mitochondrial membrane potential, so clarification is needed.

Authors: This raises an important point. Indeed, inversely, the alkalization of the cytosol (higher pH_i) decreases the net ΔpH in respect to the matrix as the IMS closely follows the pH_i . As a result, the TMRM signal would decrease. As the ΔpH contributes 20-30% to the proton gradient (Δp), lowered ΔpH is known to be compensated for by increasing NADH/FADH₂ oxidation, ion channel regulation (K⁺ uniporters, Ca²⁺ influx), proton pumping and reduction in passive proton conductance as the chemical ΔpH component diminishes (decreased leak) and adapt mitochondrial dynamics by content mixing (Santo-Domingo & Demareux, 2012; doi:10.1085/jgp.201110767. As TMRM is predominantly a measure of the $\Delta\psi_m$, these compensatory mechanisms increase the overall signal. Our respiration experiments confirm these compensatory mechanisms as i) ACT-treated cells uncouple to preserve $\Delta\psi_m$ over ATP generation and that ii) complex iv remains highly active even under ACT treatment in the KO.

Minor points:

1. Figure S3A and lines 258-260—azide is not a V-ATPase inhibitor in cells. In reference 29, it was only shown to inhibit proton pumping of the reconstituted clathrin-coated vesicle ATPase. I would omit this inhibitor.

Authors: Based on the reviewer's suggestion, we have omitted this data from Figure S3A.

2. The legends of Figure 4C-H do not match the figure.

Authors: We have corrected this.

3. Lines 560-562 and Figure 5E and F—it doesn't look like there is any significant change in mitochondrial membrane potential in the U2OS cells (unless there is something missing in Figure 5E) and the difference is very modest in Figure 5F. It might be good to soften the description of these data.

Authors: We have corrected this by softening the description. In the first submission, we used independent experiments (n=3 in the respective figures) for statistical calculations instead of individual cells (> 500 per condition).

4. Lines 671-672—the “NAD⁺-mediated acidification” is used somewhat out of context here relative to the original work. This could be confusing.

Authors: We have omitted this literature-induced reference.

5. Lines 679-680—the mutant's resistance to EtBr is consistent with its ability to maintain mitochondrial membrane potential but does not really provide any additional information about ATP synthase or signaling pathways.

Authors: We agree with this comment and have omitted this reference to shorten the discussion. We found it of particular interest that the unicellular parasites of the trypanosome family causing sleeping sickness in cattle treated with EtBr analogs become resistant upon loss of v-ATPase subunits, resistance that we also observe in our v-ATPase mutant.

Reviewer #3 (Remarks to the Author):

We appreciate the authors' substantial revisions and efforts to improve the manuscript. However, several of our concerns remain insufficiently addressed. Additionally, the authors did not clearly indicate where changes were made in the revised manuscript, which complicates the review process. For clarity and efficiency, specific changes should be referenced ideally in the rebuttal letter (e.g., “We have added a new explanation on page XXX in the revised manuscript”). I believe that the rebuttal letter serves as a crucial platform for constructive communication with reviewers. Providing clear, thorough, and well-reasoned responses not only addresses the reviewers' concerns effectively but also contributes significantly to improving the quality and clarity of the

manuscript.

Authors: We appreciate the reviewer acknowledging our substantial revisions based on their constructive comments. We sincerely apologize for the inconvenience caused by complicating the revision process by not clearly indicating and accurately referencing alterations in the responses to the reviewers. We ask to excuse the complication of referring to the previous version due to substantial revisions, which were being made changing the entire figure order (e.g., addition of Figure 3). We would like to point out that the previous submission contained both a highlighted and non-highlighted document, with the highlighted indicating all alterations made to the previous version. We agree that the exact line referencing would have simplified the revision process and again kindly ask to excuse the inconvenience thereby caused. We clearly indicate any alterations made in the current revision of the manuscript clearly in the point-to-point response and main text by highlighting and referring to the respective line in the manuscript. We also fully acknowledge that our lack of referencing has led to misunderstandings by the reviewer (e.g., removal of data), which we will try to fully amend and clarify in the responses below.

While we acknowledge the authors' completion of several experiments suggested by other reviewers, the overall flow of the manuscript has become more difficult to follow. Strengthening the logical connections between results would improve the clarity and coherence of the narrative. Should the authors revise the manuscript again, we recommend addressing the following points:

Authors: In the revised version, we have further aimed to improve coherence and clarity of our results. We acknowledge that the study is condensed but are convinced that the revision has improved the readability.

PPP Pathway Activation Evidence: The authors' response—"Our ratios from untargeted metabolomics are a crude approximation and reflect a steady-state and no flux"—remains unclear for this reviewer. This explanation requires clarification. Also, I suggested here that the NADP/NADPH ratio should be considered, but the authors did not perform that experiment directly. Although the authors were not obliged to conduct every experiment suggested by reviewers, they should address the reviewer's concerns by explanation. Moreover, the authors should indicate where and how they have changed the manuscript.

Authors: We acknowledge that we have not justified performing the suggested NADP/NADPH ratio suggested by the reviewer to ascertain PPP activation. The metabolic ratios represent an approximation as they are not quantified in untargeted metabolomics with an internal standard. To analyse PPP activity, we have now used a single-tracer method of 1,2-¹³C₂-Glucose tracing as described by Lee et al. 1998 (doi:

10.1152/ajpendo.1998.274.5.E843, new Figure 4D). In effect, when 1,2-¹³C₂-Glucose is used as a tracer, the oxidative PPP will produce Ribulose 5-phosphate (Ri5P) labeled only in C-1 position, which can be recycled into C-1, C-2 or C-3 Fructose 6-phosphate. These are subsequently converted into single labeled triose phosphates and eventually to [3-¹³C], [2-¹³C] or [1-¹³C]-lactate (m+1). Glycolysis reactions of 1,2-¹³C₂-Glucose retain both labeled carbons and yield lactate (m+2). Thus, the ratio of lactate isotopomers (m+1) to (m+2) from 1,2-¹³C₂-Glucose can be used to approximate PPP activity or Pentose cycle (PC) with mass balance equation derived from Katz and Rognstad's model (Katz and Rognstad, 1967; doi: 10.1021/bi00859a046).

After 60 min 1,2-¹³C₂-Glucose incubation, ACT-treated WT cells had 3.2% of lactate molecules containing one ¹³C substitution (m+1) and 7.8% two ¹³C substitutions (m+2). In contrast, ACT-treated ATP6AP1^{KO} cells and

EtOH treated WT had m+1 and m+2 substitution of 2.9% and 15.9%, and 2.8% and 12.4%, respectively (Figure 1, left panel). Consequently, WT cells treated with ACT displayed slower glucose incorporation but higher PC activity, an effect that was mostly negated by ATP6AP1^{KO} (Figure R1, right panel). Our glucose tracing data shows that ATP6AP1^{KO} cells have increased glucose uptake, are insensitive to ACT-induced PPP rerouting and have a reduced overall amino acid pool, potentially as a consequence of anaplerotic TCA fueling.

Figure R1 Determination of Lactate isotopomer distribution (left panel) and subsequent Pentose cycle activity (right panel) using 1,2-¹³C₂-Glucose tracing.

Sample Consistency in Figure 4C: The authors attributed the removal of certain data to a “typo” without providing an adequate explanation. They should explain what specific errors occurred and justify the data removal, as this is crucial for transparency in the review process.

Authors: In our previous submission the reviewer kindly indicated that for the treatment combination of concA and ACT in WT lacked a clear labelling of “WT” (previous Figure 4C lower two panels) - an essential attribute for the comparison of WT and ATP6AP1Ko of course (previous Figure 4C upper two panels). **Incidentally, Figure 4C did contain a typo in the previous submission as the labelling for media was missing “ACT” for the combined treatment “conca ACT” (Figure 4C lower right panel).**

No data was omitted. In the previous submission, we moved all data concerning pharmacological inhibition of the v-ATPase (including the previous Figure 4C lower panels the reviewer is referring to) to improve the flow of the manuscript. Figure 4C was split: the KO data is in Figure S6E and the concA data is in Figure S7F and has not been removed. As suggested by the reviewers, we did now clearly indicate that Figure S7F also is indeed “WT” cells and apologize for the inconvenience and are convinced this rectifies our misunderstanding of their comment. Importantly, we now understood that the reviewer pointed out that the dataset “WT ACT” from the previous Figure 4C upper panels is to corroborate the “WT” ACT from the concA dataset– in brief the reviewers indicates that these data from different experiments should yield the same results. In the previous submission, we have clearly displayed these datasets side-by-side and showing that the overall key metabolic alterations are reproducible and consistent even between independently performed datasets considering experimental-dependent variations and the technical limitations of untargeted metabolomics the reviewer is familiar with. We have displayed the data in question below side-by-side again and again (Figure R2), we kindly ask to excuse our

misunderstanding of their comment.

Figure R2 Heatmaps of amino acid levels from untargeted metabolomics.

Left panel: Heatmap of average amino acid level normalized to untreated control in cells and media from ATP6AP1^{KO}; untargeted metabolomics, mean of biological sextuplicates.

Right panel: Heatmap of average amino acid level normalized to untreated control in WT cells and media from conoA +/- ACT; untargeted metabolomics, mean of biological sextuplicates; 3-days treatment.

LAMP2 Expression in Figure 4F: The authors stated that they “changed the text to emphasize this” but failed to specify how or where these changes were made. A clear indication of these modifications within the manuscript is necessary.

Authors: We apologize for the inconvenience and indicate this clearly in the resubmission in line 293-95:

“Lysosome-associated membrane glycoprotein 1 and 2 (LAMP1/2) showed comparable levels but were retained higher in the gel in ATP6AP1KO cells.”

Lysotracker Signal Interpretation in Figures 4G and 4H: Although new experiments were performed, my question is “either an increase in the number/volume of acidic vesicles or a decrease in lysosomal pH, or both” “discussing the metabolic implications of these changes”. Please explain this point.

Authors: Our extended experimentation shows that this increased signal is an increase in the overall volume of acidic vesicles as determined by Lysotracker DND-99. Our mCherry-LAMP1-GFP experiments determine that the lysosomal fraction of these acidic vesicles shows “alkalinization”. We provide metabolic explanation based on our new glucose tracing experiments as discussed above and, in the manuscript, (new Figure 4; 384-397; discussion: 536-543). We have discussed effects of lysosomal size alterations modifying hydrolase-assisted macromolecule degradation as well as regulation of osmolytes consisting of ions and organic solutes (and water). Lysosomal volume increase upon dysfunction (e.g., macromolecule accumulation) is a commonly observed phenomenon (Hu et al., 2022 do: 10.1083/jcb.202109133). Multiple lysosomal channels and transporters (H⁺, Na⁺, K⁺, Ca⁺, Cl⁻) exist, whose functions might be influenced by altered lysosomal size and in turn affect both trafficking and solute flux beneficial in the pathological state of loss/leakage (leakage) of the mitochondrial membrane potential.

Intracellular pH and Mitochondrial Membrane Potential: The authors employed a method to quantify lysosomal pH and intracellular pH, allowing more detailed measurement of pH changes. While this is a great attempt, it raises a new question. In a previous manuscript, the authors suggested that v-Atpase inhibition by ATP6AP1KO or ConcA causes intracellular acidification, which increases mitochondrial membrane potential and improves cell fitness under ACT treatment (Previous manuscript, Page21 line 500-501 “These data suggest that intracellular acidification through inhibition of v-ATPase function improves cell fitness under mitochondrial dysfunction”). However, in the revised manuscript, making the intracellular environment more alkaline (or near neutral) to improve cell fitness is important. For example, Fig. 3H in the revised manuscript suggests that ACT treatment shifts the intracellular environment to acidic conditions, while ATP6AP1KO-ACT treatment conditions return it to the alkaline (neutral) side. These results are completely different from the claim of the previous manuscript. Particularly uncomfortable is that Fig. 5H in the previous manuscript shows a significant increase in intracellular pH in ATP6AP1KO compared to WT, however in the revised manuscript, Fig. 3H, there is no apparent difference between WT and ATP6AP1KO in intracellular pH. These results should be the same conditions, so an explanation is required as to why this difference occurred. In addition, the claims in the revised manuscript do not explain how the loss of ATP6AP1 ameliorates the reduction in cell fitness caused under ACT treatment.

Authors: We agree with the reviewer's comment regarding altered pH. Our initial observations were based on steady-state measurements using the commonly used cytosolic dye pHrodo green. Our initial experiments were performed without standard curves and did not test the effect of actinonin on intracellular pH. In our revised version, we used pH-calibrated standard curves with defined buffers and the use of nigericin/valinomycin-mediated equilibration of the plasma membrane for WT and ATP6AP1 KO independently for every condition to accurately assess their respective pH, instead of reporting mean fluorescence – as kindly suggested by the reviewer – to correct for confounding factors such as dye loading variability, leakage and cell thickness (Casey et al., 2010; doi: 10.1038/nrm2820). We show that v-ATPase inhibition alters cytosolic pH and how metabolism is rerouted by metabolic tracing experiments beneficial to cell fitness under mitochondrial dysfunction (Figure 4). In the initial version of the manuscript, we used pHrodo Green to determine intracellular pH. However, as a non-ratiometric dye, pHrodo Green primarily provides qualitative information, with fluorescence increasing as pH decreases. While this makes it useful for detecting acidification events such as phagosome maturation, it is less suitable for accurate, quantitative measurement of cytosolic pH. Following consultation with international experts in intracellular pH measurement, we have now adopted the use of BCECF, a well-established ratiometric dye. BCECF allows more precise and quantitative assessment of pH, as its dual-excitation ratio is independent of dye concentration, photobleaching, and cell thickness.

We apologize if this change gives the impression that the conclusions of our cytosolic pH experiments have shifted significantly during the review process. Rather, we would like to emphasize that this refinement reflects our commitment to increasing the robustness and reliability of the data, in line with the goals of peer review.

Reviewer #4 (Remarks to the Author):

Reviewer #5 (Remarks to the Author):

The revised manuscript has improved, though its complexity remains high, but the authors have largely addressed this Reviewer's comments. However, this Reviewer would agree that particularly comments by Reviewers #2 and #3 need to be addressed.

Reviewer #6 (Remarks to the Author):

REVIEWERS' COMMENTS

We sincerely thank the reviewers for their valuable time and thoughtful feedback, which has significantly strengthened and refined our study.

Reviewer #1 (Remarks to the Author):

In the revised manuscript, the authors adequately addressed my concerns.

We thank the reviewer for their feedback.

Reviewer #2 (Remarks to the Author)

This revised version of the manuscript is written more clearly and has satisfied most of my previous comments. The major message of the paper is that partial inhibition of the lysosomal V-ATPase can compensate for mitochondrial defects arising from mistranslation, and the authors have proven this point. However, the data indicate that this compensation depends on a relatively small increase in lysosomal acidification and there are still some places where the wording does not convey this. I would recommend only the following text changes:

1). Lines 353-354, 406 and 559: Change "V-ATPase deficient" and "V-ATPase disruption" to partial loss of V-ATPase function or partially disrupted V-ATPase function.

We thank the reviewer for their feedback and have altered the respective passages for a unified nomenclature.

2). The use of "low V-ATPase inhibition" and "low lysosomal defect" on lines 462, 464, 504-505, 530, 538-539, 543, 546, 549 are potentially confusing. I would recommend "partial V-ATPase inhibition" or "partial lysosomal defect" because this appears to be what is happening with very low concentrations of concanamycin/bafilomycin or ATP6AP1 KO.

We thank the reviewer for their feedback and have altered the respective passages for a unified nomenclature.

3). Lines 521-523 which are in a revised section of the discussion indicate that "enlarged, less acidic lysosomes promote a shift to catabolism...during energy stress". However, recent data indicates that energy stress can also induce a shift to more perinuclear and more acidic lysosomes (Ebner et al. (2023) Cell 186:5328). This reference is also relevant here.

We included this recent data.

Reviewer #3 (Remarks to the Author)

In this revision, the authors have added quantitative experiments using glucose tracing and BCECF-AM, which significantly enhance the reliability of the data. As a result, some

interpretations differ from those in the original manuscript; however, the central concept of “extra-organelle buffering” is now more robust and convincing, supported by substantial data. The manuscript’s structure has also been simplified and clarified, improving readability.

We thank the reviewer for their feedback and suggestions.

I believe no additional experiments are necessary. Nevertheless, to aid readers' understanding, some sections require explanation or clarification as described below.

Minor points to be clarified

1. Regarding the activation of the PPP pathway, I understand that the authors did not examine the NADP/NADPH ratio due to concerns about quantitative reliability. Instead, in the new experiments presented in Figure 4D, glucose tracing using C¹³ was performed. This experiment is elegant, but the explanation is somewhat limited. While the manuscript clearly describes glucose uptake and lactate synthesis via the PPP pathway, it is unclear how TCA intermediates and amino acids (AAs) were quantified. The y-axis in the figure is not fully explained. Does it represent relative abundance, fraction labeled, or another measure? This information should be explicitly stated, either in the figure legend or in the main text. The “Methods” section details the experimental procedures but does not describe the calculation methods.

We have specified the determination of relative abundance of TCA intermediates and amino acids (AAs) in the respective passage and extended and detailed the experimental method section.

2. Additionally, rather than combining all these results into a single Figure 4D, it may be preferable to present them as separate panels (Figures 4D, E, F, G, and H) with detailed explanations in the main text.

We have aimed to be more precise in the description of the ¹³C tracing experiment. In order for the reader to understand that Figure 4D is a single experiment and to set it apart from the other metabolomic analyses, we opted to not separate it into single panels. We hope that this is an agreeable option.

3. Regarding the glucose tracing figure, it appears that the red and blue circles both represent C¹³ labeling. If so, is there a specific reason for using different colors? Initially, it may be interpreted as indicating distinct molecules. Clarifying this in the legend—for example, with a note such as “filled circle indicates C¹³ position in glucose”—would be helpful.

We agree with the reviewer's comment and have clarified the meaning of the colours to represent the metabolic pathways with filled circles indicating ¹³C-position in glucose and colour the derivation from glycolysis or PPP combustion.

4. In Figure 4D, for ATP6AP1 KO, “KO” should be written as a superscript.

We have corrected this mistake.

5. Are GADP and G3P distinct substances? If so, it would be helpful to clarify their identities.

We have now specified all abbreviations in the figure legend.

Reviewer #4 (Remarks to the Author):
